# CAMx-UNIPAR Simulation of SOA Mass Formed from Multiphase Reactions of Hydrocarbons under the Central Valley Urban Atmospheres of California

Yujin Jo[1], Myoseon Jang[1], Sanghee Han[1], Azad Madhu[1], Bonyoung Koo[2], Yiqin Jia[2], Zechen Yu[1], Soontae Kim[3], and Jinsoo Park[4]

[1]Department of Environmental Engineering Sciences, Engineering School of Sustainable Infrastructure and Environment, University of Florida, Gainesville, FL, USA.
[2] Bay Area Air Quality Management District, San Francisco, CA, USA.
[3] Department of Environmental and Safety Engineering, Ajou University, Suwon, South Korea.
[4]Air Quality Research Division, National Institute of Environmental Research, Environmental Research Complex, Incheon, South Korea.

*Correspondence to*: Myoseon Jang (mjang@ufl.edu)

**Abstract.** The UNIfied Partitioning-Aerosol phase Reaction (UNIPAR) model was integrated into the Comprehensive Air quality Model with extensions (CAMx) to process Secondary Organic Aerosol (SOA) formation by capturing multiphase reactions of hydrocarbons (HCs) in regional scales. SOA growth was simulated using a wide range of anthropogenic HCs, including ten aromatics and linear alkanes with different carbon-lengths. The atmospheric processes of biogenic HCs (isoprene, terpenes, and sesquiterpene) were simulated for major oxidation paths (ozone,

OH radicals, and nitrate radicals) to predict day and night SOA formation. The UNIPAR model streamlined the multiphase partitioning of the lumping species originating from semi-explicitly predicted gas products and their heterogeneous chemistry to form non-volatile oligomeric species in both organic aerosol and inorganic aqueous phase. The CAMx–UNIPAR model predicted SOA formation at four ground urban sites (San Jose, Sacramento, Fresno, and Bakersfield) in California, United States during wintertime 2018. Overall, the simulated mass concentrations of the

total organic matter, consisting of primary organic aerosol and SOA, showed a good agreement with the observations. The simulated SOA mass in the urban areas of California was predominated by alkane and terpene oxidation products. During the daytime, low-volatility products originating from the autoxidation of long-chain alkanes considerably contributed to the SOA mass. In contrast, a significant amount of nighttime SOA was produced by the reaction of terpene with ozone or nitrate radicals. The spatial distributions of anthropogenic SOA associated with aromatic and

alkane HCs were noticeably affected by the southward wind direction, owing to the relatively long lifetime of their atmospheric oxidation, whereas those of biogenic SOA were nearly insensitive to wind direction. During wintertime 2018, the impact of inorganic aerosol hygroscopicity on the total SOA budget was not evident because of the small contribution of aromatic and isoprene products, which are hydrophilic and reactive in the inorganic aqueous phase. However, an increased isoprene SOA mass was predicted during the wet periods, although its contribution to the total

SOA was little.

## 1 Introduction

Secondary Organic Aerosol (SOA) contributes a significant portion of the Particulate Matter (PM) burden globally, with adverse impacts on health (Pye et al., 2021), visibility (Liu et al., 2022), and climate (Hallquist et al., 2009; Mahrt et al., 2022). SOA forms via gas–particle partitioning (Odum et al., 1996) of oxygenated products formed from the photooxidation of various hydrocarbons (HCs). Additionally, heterogeneous chemistry of oxidized carbons in the aerosol phase significantly contributes to SOA growth by forming non-volatile oligomeric mass (Hallquist et al., 2009; Kalberer et al., 2004; Tolocka et al., 2004). In particular, the contribution of heterogeneous chemistry to SOA mass increases with the aerosol water content associated with the hygroscopicity of electrolytic inorganic aerosol (Volkamer et al., 2009) and aerosol acidity (Jang et al., 2002).

SOA models have been used at regional and global scales to predict the contribution of SOA mass to the total atmospheric organic aerosol (Zhang et al., 2021; Hodzic et al., 2015). The prediction of SOA formation has been approached by a conventional gas–particle partitioning of surrogate products (Odum et al., 1996) or the volatility basis set (VBS) approach (Donahue et al., 2006). However, these partitioning-based models poorly integrate the heterogeneous reactions of multifunctional organic products. SOA models include isoprene but have no features for aqueous reactions of products originating from various HCs (Vasilakos et al., 2021; Yu et al., 2022). Evidently, the limitation in the heterogeneous chemistry of various products can lead to deviations in the predicted SOA mass concentration from observations (Baker et al., 2015; Hayes et al., 2013; Carlton and Baker, 2011). For example, SOA formation has been underestimated in highly humid regions, such as the Southeastern United States (US) in summer, when the hygroscopic inorganic aerosol is wet and aqueous reactions of reactive organic species are increased (Zhang et al., 2018). To improve the prediction of SOA, SOA models have employed large partitioning-based parameters, gas mechanisms to form low-volatile products via autoxidation (Mayorga et al., 2022; Jokinen et al., 2015; Pye et al., 2019), or nighttime oxidations of biogenic HCs with nitrate radical and ozone (Zaveri et al., 2020; Gao et al., 2019). These model parameters and reactions can improve SOA predictions, but they can lead overprediction of SOA under certain situations. For example, Liu et al. (2021) reported in their regional simulation using the Community Earth System Model (CESM2.1) that the model parameters in the VBS model were too high for terpenes and led to overprediction of SOA mass with biased parameters that can overpredict organic matter (OM) mass.

The UNIfied Partitioning-Aerosol phase Reaction (UNIPAR) model predicts SOA mass by employing the multiphase partitioning of organic species between gas, organic and inorganic phases with improved prediction of thermodynamic parameters (i.e., activity coefficients of organic species in the salted inorganic phase under various humidity conditions) and in-particle chemistry (Beardsley and Jang, 2016; Im et al., 2014; Yu et al., 2021c; Zhou et al., 2019). Chemical species originating from explicit gas mechanisms are lumped based on volatility and reactivity in the aerosol phase. The physicochemical parameters of lumping species allow the UNIPAR model to process multiphase partitioning and in-particle reactions (i.e., oligomerization and acid catalyzed reactions). The model parameters and equations in UNIPAR have been demonstrated for various SOA originating from a wide range of HCs such as isoprene, terpenes, aromatics, and gasoline and demonstrated through chamber data under different levels of $NO_x$ and different types of inorganic aerosols associated with aerosol's acidity, hygroscopicity, and phase states (i.e., dry and wet) (Im et al., 2014; Beardsley and Jang, 2016; Cao and Jang, 2010; Zhou et al., 2019; Yu et al., 2021c; Han and Jang, 2022).

In a recent study by Yu et al. (2022), the UNIPAR model was integrated with the Comprehensive Air Quality Model with extensions (CAMx) (Environ, 2020) and demonstrated its ability to predict SOA mass for South Korea during the Korea-United States Air Quality (2016 KORUS-AQ) campaign. The CAMx–UNIPAR model well predicted SOA formation through the processing of aqueous phase reactions of organics and their gas-aqueous partitioning during wet periods during the KORUS-AQ campaign. Although the OM mass predicted with CAMx–UNIPAR was noticeably better than that of the CAMx model with the conventional partitioning-based two-product model, the OM mass was somewhat underestimated compared with the observations. The underestimated OM was possibly due to missing precursors and missing chemistry in both the gas and aerosol phases.

In this study, the CAMx–UNIPAR model has been updated to include the SOA formation from long-chain alkanes and nighttime chemistry of biogenic HCs. Long-chain alkanes are regarded as essential precursors for SOA formation (Aumont et al., 2012; Madhu et al., 2022). Madhu et al. (2022) have recently added an autoxidation mechanism into alkane semi-explicit oxidation mechanisms, improving the predictability of alkane SOA using the UNIPAR model against their chamber study. The resulting alkane model parameters have been newly implemented into CAMx–UNIPAR. In addition, the UNIPAR model of this study has been expanded to simulate biogenic SOA based on three major oxidation paths (i.e., OH radicals, ozone and nitrate radicals) being capable of nighttime SOA formation that is dominated by oxidation with ozone and nitrate radicals (Han and Jang, 2023). The performance of the extended CAMx–UNIPAR model has been demonstrated to predict the SOA budget under urban environments in the Central Valley, California (CA), US during the winter (January–February) in 2018. The impact of aerosol hygroscopicity on SOA formation under varied humidity conditions, spatial distributions of each SOA species, and diurnal patterns of SOA formation associated with nighttime oxidation of biogenic HCs were characterized using the CAMx–UNIPAR simulation in California's urban environments.

## 2 Methods

### 2.1 The UNIPAR SOA model

The structure of the UNIPAR SOA model has been described previously (Cao and Jang, 2010; Im et al., 2014; Beardsley and Jang, 2016; Zhou et al., 2019; Yu et al., 2021c; Yu et al., 2022; Choi and Jang, 2022; Madhu et al., 2022; Han and Jang, 2023). The CAMx–UNIPAR of this study simulates SOA formation via multiphase reactions of ten aromatics, three biogenics (isoprene, terpene, and sesquiterpene), and three alkane groups as summarized in Table 1. The lumping species, which are generated by using semi-explicitly predicted products of the oxidation of HCs, are involved in multiphase partitioning and aerosol-phase reactions to form SOA mass.

The prediction of stoichiometric coefficients associated with the lumping species from the oxidation of ten aromatics (MCM v3.3.1(Jenkin, 2004)) potentially increases the accuracy of the prediction of aromatic SOA formation under varying emissions. For biogenic HCs, recently identified biogenic oxidation mechanisms that yielded low-volatility products have been applied to the prediction of lumping species. For example, the peroxy radical autoxidation mechanism (Roldin et al., 2019) is known to form highly oxygenated organic molecules (HOMs) (Molteni et al., 2019). Additionally, HOMs form via the accretion reaction to form ROOR from $RO_2$ species (Bates et al., 2022; Zhao et al.,

2021). These HOMs are included in biogenic lumping species in the current CAMx–UNIPAR. In order to represent day and night chemistry, CAMx–UNIPAR is equipped with the oxidation-path-dependent stoichiometric coefficients from individually processed reactions of biogenic HCs with three major oxidants (OH radicals, ozone, and nitrate radicals) (Han and Jang, 2023). Thus, nighttime SOA formation, which is dominated by oxidation via ozone and nitrate radicals, can be simulated in CAMx–UNIPAR. By using the recent chamber study by Madhu et al. (2022), the lumping species generated from the oxidation of long-chain alkanes raging from C9 to C24 have been newly integrated with CAMx–UNIPAR, which increases the predictability of SOA formation from intermediate volatility organic compounds (IVOCs). Alkane gas mechanisms include autoxidation of alkoxy radicals to form HOMs (Madhu et al., 2022; Crounse et al., 2013; Bianchi et al., 2019; Roldin et al., 2019). Importantly, the estimation of activity coefficients of lumping species in both the organic phase and the inorganic aqueous-phase enables the simulation of multiphase partitioning of lumping species and their aerosol phase reactions. Aqueous reactions in CAM-UNIPAR facilitate the evaluation of the impact of humidity, aerosol acidity, and aerosol-phase state on SOA formation in regional scales.

Briefly, the key components of the UNIPAR model are described as follows:

1) The SOA mass forms via multiphase partitioning, organic-phase oligomerization, and aqueous reactions in a wet inorganic salted solution. For the multiphase SOA formation, a fundamental assumption is that organic constituents are internally mixed with inorganic constituents under the liquid-liquid phase separation between organic phase and salted aqueous phase.

2) As shown in Table 1, the 251 lumping groups in the current UNIPAR cover oxygenated products predicted from explicit gas mechanisms (MCM v3.3.1(Jenkin, 2004)) of 16 precursors (ten aromatics, three alkane groups in different carbon-lengths, isoprene, terpene, and sesquiterpene) with three major oxidation paths (OH radicals, ozone, and nitrate radicals) to represent HC oxidation in day and night both. Pre-determined mathematical equations dynamically build the stoichiometric coefficient arrays, which are connected to volatility-reactivity based 251 lumping species and applied to gas-particle partitioning and heterogeneous reactions to from SOA. The stoichiometric coefficient array reflects the influence of $NO_x$ levels and gas aging on gas-product distributions based on $RO_2$ chemistry between $RO_2+NO$ reactions and $RO_2+HO_2$ reactions under varying $NO_x$ conditions during day and night.

3) Lumping species' physicochemical parameters, such as molecular weight ($MW_i$), oxygen-to-carbon ratio ($O:C_i$), and hydrogen bonding ($HB_i$), are used to determine their multiphase partitioning. Each precursor group uses a single set of physicochemical parameter arrays associated with lumping species: for example, 50 arrays for aromatics, 50 arrays for alkanes, 50 arrays for terpene, 50 arrays for sesquiterpene, and 51 arrays for isoprene as shown in Table 1. The physicochemical parameters of lumping species for $MW_i$, $HB_i$, and the $O:C_i$ ratios are reported in Tables S1–S3, respectively.

4) The concentration of lumping species is distributed into gas ($C_g$), organic ($C_{or}$), and inorganic phases ($C_{in}$) using partitioning coefficients estimated based on Pankow's absorptive partitioning model (Pankow, 1994) with vapor pressure, the estimated activity coefficients of lumping species in both the organic and inorganic phases (Zhou et al., 2019; Yu et al., 2021c; Madhu et al., 2022; Han and Jang, 2022; Han and Jang, 2023), and aerosol's average molecular weight in each phase.

5) Aerosol phase reaction rate constants of lumping species in the organic phase and inorganic phase are calculated with kinetic parameters, such as lumping species' reactivity scales (Table S4) and their basicity constants (Table S5). The kinetic parameters used in CAMx–UNIPAR are updated by removing the artifact from gas-wall partitioning (Han and Jang, 2020; Han and Jang, 2022).

6) Heterogeneously formed OM (OMH), which is produced via oligomerization in the organic phase and the inorganic phase, is treated as non-volatile OM. The impact of viscosity on aerosol growth is also considered by including the equation term into aerosol-phase rate constants as a function of the average molecular weight of OM and the O:C ratio (Han and Jang, 2022). Aqueous reactions in the presence of wet-inorganic aerosol are linked to acid-catalyzed reactions, which are processed under broad ranges of aerosol acidity ($[H^+]$) and relative humidity (RH) levels to form both dry and wet inorganic aerosols. The species in the lowest volatility group, which are multifunctional and dominantly present in aerosol phase, can easily react in aerosol phase via various unidentified reactions (esterification and oxidations) and form non-volatile species. Thus, the lumping species in the lowest volatility group are involved in oligomerization with a high reaction rate constant used for glyoxal, regardless of lumping groups' reactivity scale.

7) The SOA mass in UNIPAR is estimated by gas-particle partitioning (OMP) and heterogeneous reactions (OMH) in both organic and inorganic phase. The SOA mass formed from partitioning (OMP) is estimated using the Newtonian method (Schell et al., 2001) based on a mass balance of organic compounds between the gas and particle phases governed by Raoult's law. OMH is considered as a pre-existing absorbing organic material for gas-particle partitioning (Cao and Jang, 2010; Im et al., 2014). The resulting OMP is updated by the addition of $C_{in}$.

8) To treat SOA formation in the inorganic aqueous phase, inorganic composition and aerosol acidity are predicted using an inorganic thermodynamic model, ISORROPIA (Fountoukis and Nenes, 2007), and then incorporated into the UNIPAR model. Mutual deliquescence relative humidity (MDRH) is predicted from the ISORROPIA model. The efflorescence relative humidity (ERH) is predicted using a pre-trained neural network model based on the inorganic composition (Yu et al., 2021b). The aerosol state is determined to be wet (organic phase + inorganic aqueous phase) or dry (organic phase + solid-dry inorganic phase) with MDRH and ERH.

9) The formation of organosulfate in the form of dialkylsulfate, non-electrolytic and neutral ester of sulfuric acid with organic species, is reported in numerous laboratory studies (Liggio et al., 2005; Surratt et al., 2007; Li et al., 2015). In the model, the formation of dialkylsulfate is simulated by using the Hinshelwood-type reaction (Im et al., 2014). The decreased acidic sulfate due to the dialkylsulfate formation is applied to inorganic thermodynamic model (ISORROPIA) to calculate aerosol acidity and aerosol water content for the next step.

**2.2 The CAMx–UNIPAR integration**

Fig. 1 shows the overall structure of the CAMx–UNIPAR model. The Weather Research and Forecast model (WRF v4.1) generated meteorological input data to drive the CAMx v7.1 using the North American Mesoscale (NAM) data of the National Centers for Environmental Prediction (NCEP). The WRF modeling domains comprised three nested grids with horizontal resolutions of 36, 12, and 4km. The outer domain, with a 36km horizontal resolution, covered the entire western US. The second domain, with a 12km horizontal resolution, covered California and portions of Nevada. The third domain, with a 4km horizontal resolution, covered central California. The air quality simulation for

gas and particle pollutants in central California was performed for the third domain (Fig. 2a) with 185×185 grids and

22 vertical layers. The detailed configurations of the WRF and CAMx modeling system are described in Table S6. Emission inputs are based on California Air Resources Board regional inventories and provided by the Bay Area Air Quality Management District (BAAQMD, 2023).

The UNIPAR SOA model was incorporated with the CAMx model as a sub-model to predict SOA formation in

California. Precursor HC consumption and other gas species were estimated using the SAPRC07TC gas mechanism (Hutzell et al., 2012) in the CAMx model. The inorganic thermodynamic model (ISORROPIA) provides the inorganic aerosol composition at equilibrium for inorganic aerosol species. The current CAMx–UNIPAR model includes 251 lumping species, of which 50 originate from ten aromatics; 50 originate from three alkane groups; 50 originate from terpene; 50 originate from sesquiterpene; and 51 originate from isoprene (Table 1).

The emissions of ten aromatic HCs were sourced from benzene, toluene, three xylenes, and 1,2,4-trimethylbenzene. Toluene was used to estimate the emissions of ethylbenzene and propylbenzene using factor of 0.082, and 1,2,4-trimethylbenzene was used to estimate the emissions of 1,2,3-trimethylbenzene and 1,3,5-trimethylbenzene using factor of 0.3.  Long chain alkanes are important IVOCs, which are emitted from automobile exhaust (Pye and Pouliot, 2012; Ensberg et al., 2014) and plants (Simoneit, 2002; Yao et al., 2009; Li et al., 2022). Laboratory studies showed

that SOA yields from alkanes changed with carbon length (Lim and Ziemann, 2009; Tkacik et al., 2012; Presto et al., 2010; Srivastava et al., 2022). In the model, alkanes in different carbon-chain lengths are split into three groups due to their different SOA formation potential: C9-C14, C15-C19, and C20-C24 (Table 1). Speciation factor of 0.52 was used to estimate the emission of alkanes from SAPRC07 ALK5 species, which contains other non-aromatic compounds (i.e., alcohols, ethers, esters, and carboxylic acids) in addition to alkanes (Carter, 2015). The emission

factor of alkanes that were not fully covered by ALK5 (i.e., alkanes longer than C19, (Carter, 2015)) was estimated using the HC emission recently reported by the US Environmental Protection Agency (US EPA) (Pye et al., 2022) and the study by Mcdonald et al. (2018). In this study, all types of alkanes including linear, branched, and cyclic alkanes were surrogated with linear alkanes based on a carbon number.

The SOA mass in the CAMx–UNIPAR model is attributed to the OMH and OMP for each precursor (Table 1). The

total SOA is the sum of OMH_AR, OMH_AK, OMH_TE, OMH_SP, OMH_IS, OMP_AR, OMP_AK, OMP_TE, OMP_SP, and OMP_IS. Total OM is the sum of total SOA and single primary organic aerosol (POA) species. POA is a single non-volatile species that does not chemically evolve (Environ, 2020). The simulation period was the winter (January–February) in 2018, excluding the wildfire season in California. Here, we focused on the simulation results that showed considerable OM levels between 01/23/2018 and 02/24/2018.

**2.3 Observation sites in California**

The US EPA has operated the national Chemical Speciation Monitoring Network (CSN) since 2000 to obtain information on the spatial and temporal chemical composition of ambient fine particles (Solomon et al., 2014). CSN measures the major components of fine particles that are 2.5 microns or less in diameter ($PM_{2.5}$) including sulfate, nitrate, ammonium, organic carbon (OC) and elemental carbon (EC) and operates on a 1-in-3-day sample collection

schedule.  Fig. 2(a) shows four CSN sites in California - San Jose, Sacramento, Fresno, and Bakersfield selected in

this study. Table 2 summarizes information on the observation sites. The San Jose site is within the boundaries of the San Francisco Bay Area Air Basin (SFBAAB or Bay Area). The Sacramento, Fresno, and Bakersfield sites are located on flat land in the Central Valley. The Central Valley is bounded by mountain ranges and coastal areas. This bowl-shaped geography of the Central Valley creates conditions conducive to holding pollutants in place and the accumulation of secondary photochemical compounds.


The Great Basin High occurred in California during the cold season (Herner et al., 2005). When the Great Basin High develops, the pressure gradients between the coast and the Central Valley decline, and the Planetary Boundary Layer (PBL) height becomes low. This condition holds pollutants close to the ground, allowing them to accumulate over several days, as fine particles and their precursors continue to be emitted. Due to a combination of unfavorable geographical and meteorological factors for ventilation, many cities in the bowl-shaped region, including San Jose, Sacramento, Fresno, and Bakersfield, undergo persistent air quality issues (Lurmann et al., 2006; Young et al., 2016; Prabhakar et al., 2017; Chen et al., 2020; Sun et al., 2022).


### 2.4 Organic Carbon and Elemental Carbon data

The measured mass concentrations of EC at the San Jose, Sacramento, Fresno, and Bakersfield sites during the simulation period were used to separate the concentration of the measured total OC ($OC_{tot}$) into primary OC (POC) and secondary OC (SOC) (Turpin and Huntzicker, 1995). Because EC and POC often have the same sources, there is a representative ratio of POC/EC for a given area (Strader et al., 1999). The POC is calculated based on Eq. 1 and the SOC is estimated by subtracting the resulting POC from the $OC_{tot}$ as shown in Eq. 2 (Turpin and Huntzicker, 1995).


$$POC = EC \times \left(\frac{POC}{EC}\right) \tag{1}$$

$$SOC = OC_{tot} - POC \tag{2}$$


EC and $OC_{tot}$ were measured from the same CSN system. In this study, the POC/EC ratio reported by Strader et al. (1999) by using field data obtained in California during the winter of 1995–1996 has been applied to estimate POC and SOC. The averaged POC/EC ratio was set to 2.4 with an upper limit of 2.75 and a lower limit of 2.05. $OC_{tot}$ data, the estimated POC, and the estimated SOC were converted to total OM, POA, and SOA, respectively, using a conversion factor of 1.7 (Mcdonald et al., 2015).


### 2.5 Emissions in California

To show the overall tendency, emissions of HCs and POA were averaged between January and February in 2018 as shown in Fig. 2. HC emissions are speciated according to SAPRC07 gas mechanisms. The amounts of major SOA precursors - aromatics (ARO), alkane (ALK5), terpene (TERP), sesquiterpene (SESQ), and isoprene (ISOP) emissions for each site and the total of the domain are shown in Fig. 2(b). The spatial distributions of POA and major SOA precursors emissions for California are shown in Fig. 2(c–h). Aromatic emission is the sum of benzene, toluene, *o*-xylene, *p*-xylene, *m*-xylene, and 1,2,4-trimethylbenzene. The alkane corresponds to ALK5 in SAPRC07 emission data. On average, terpene, isoprene, alkane, aromatics, and sesquiterpene emissions in California accounted for 44, 31, 18, 7, and 1%, respectively, indicating the dominance of biogenic emissions, particularly terpene emissions, in California.


For the selected sites, alkane emissions accounted for more than 60%, followed by aromatic emissions at approximately 32%. Terpene and isoprene emissions accounted for less than 5%, and sesquiterpene emission was negligible, indicating the dominance of alkane emission in urban areas. Fig. 2(c–h) depicts a distinct difference in the spatial distribution of biogenic (terpene, sesquiterpene, and isoprene) and anthropogenic (POA, aromatics, and alkane) emissions. The spatial patterns of biogenic emissions followed the distribution of mountain ranges surrounding the

Central Valley, whereas the spatial patterns of POA, aromatics, and alkane emissions were concentrated in the source area such as industrial, transportation, and buildings in the Central Valley and Bay area. In addition, long-chain alkanes can also be attributed to biogenic sources. For example, alkanes are emitted from terrestrial higher plants cuticle wax, suspended spores, and aquatic plants, and they can be emitted to atmosphere during wildfires (Simoneit, 2002; Yao et al., 2009; Li et al., 2022).

**2.6 Meteorological variables**

The time series and scatter plots of the surface temperature, relative humidity, and wind speed simulated by the WRF model against observations at the four sites are shown in Figs. S1–S3, respectively, and their statistical summaries of correlation coefficient (R), Index of Agreement (IOA), and Mean Bias (MB) are described in Table S7. Detailed descriptions of these statistical parameters are presented in Table S8. Overall, the surface temperature, relative

humidity, and wind speed were well predicted with high R and IOA values, reducing uncertainties caused by the meteorological input data in the air quality simulation. Fig. S4 shows the wind rose plots for the four sites. Northwesterly winds prevailed at all sites, leading to the convergence of the pollutants in the southern area. Fig. S5 shows the time series of the hourly simulated PBL height at the four sites. The relatively low PBL height and weak wind speed appeared together in some periods (i.e., 01/26/2018–02/13/2018), creating conditions conducive to the

accumulation of pollutants.

In the CAMx–UNIPAR model, the phase state of inorganic aerosol is determined by using the calculated ERH based on a pre-trained neural network mathematical equation (Yu et al., 2021b) and inorganic compositions (sulfate, nitrate, and ammonium) predicted from an inorganic thermodynamic model, ISORROPIA (Fountoukis and Nenes, 2007). The ERH and DRH of the salted aerosol and its phase state (wet or dry) is illustrated in Fig. S6. During the simulation

period, the inorganic aerosol phase was mostly wet, except for a few dry periods.

**3 Results and Discussion**

**3.1 Simulation of organic matter**

Fig. 3 shows the time series of the simulation of the total OM (Fig. 3a–d), POA (Fig. 3e–h), and SOA (Fig. 3i–l) with the CAMx–UNIPAR model against the observations at the four sites with statistical parameters (R, IOA, MB, root

mean square error (RMSE), and normalized MB (NMB) as described in Table S8. Overall, the simulated mass concentrations of total OM, consisting of POA and SOA, at four ground sites (San Jose, Sacramento, Fresno, and Bakersfield) showed a good agreement with the observations. The high R (0.77–0.89) and IOA values (0.82–0.87) in

Fig. 3 indicate a good agreement between the predicted total OM and observed total OM concentrations. The marginally negative values for MB and NMB indicate that the model underestimated the total OM to some extent.

The simulation of the POA and SOA could capture observations, but it shows a marginal decrease in their R and IOA values compared with the total OM. The simulated POA was slightly overpredicted and the simulated SOA was somewhat underestimated at the four urban sites. Both MB and NMB values of POA were positive, while those of SOA were negative. The negative bias of the predicted SOA is generally greater than the positive bias of the POA, which affects the negative bias of the total OM from the observations.

The underestimation of SOA mass can be attributed to missing precursor HCs and unidentified chemistry in the gas and aerosol phases. The precursor HCs such as phenols (Bruns et al., 2016; Majdi et al., 2019; Choi and Jang, 2022), branched and cyclic alkanes (Chan et al., 2013; Gentner et al., 2017; Madhu et al., 2023), and polycyclic aromatic HCs (PAHs) (i.e., naphthalene) (Riva et al., 2015; Wang et al., 2021) are currently missing in the UNIPAR model. For example, all alkanes in this study are treated with linear alkanes, increasing some uncertainties. Branched alkane

SOA can be slightly overestimated by substituting it with linear alkanes at the same carbon length, but cyclic alkane SOA can be underestimated by using linear alkanes (Madhu et al., 2022; Madhu et al., 2023). Zhang and Ying (2012) reported that PAHs including naphthalene, methylnaphthalene and dimethylnaphthalene can contribute to 4% of the anthropogenic SOA mass. Pye et al. (2022) reported in their regional simulation using the Community Regional Atmospheric Chemistry Multiphase Mechanism (CRACMM) that a significant amount of phenolic compounds is

missing in the current models. They estimated that most phenol mass (69%) is directly emitted with the balance from benzene oxidation, and cresols are mainly chemically produced (80%). The missing emissions of phenol and cresol can account for 30% of the total aromatic SOA mass (Pye et al., 2022). The SOA formation from low volatility aromatic HCs (aromatics substituted with long-chain alkyl groups) is also missing in the SOA simulation of this study. Additionally, the statistical analysis may vary, owing to the lack of observational data for the simulation periods. For

example, the OC and EC observation data were only available for daily averaged data every three days (Sect. 2.3), and this low time-resolution observational data limited the verifications of the simulated OM mass, especially SOA formation during daytime and nighttime from the CAMx–UNIPAR simulations. In addition, the observed POA estimated using the POC/EC ratio (Sect. 2.4) also has uncertainties because the POC/EC ratio is inconsistent from source to source but varies between sources (Strader et al., 1999). The simulated POA can also be inaccurate due to

the uncertainty in primary OC emission data.

The SOA/total OM mass ratios estimated from the observation data ranged from 0.50 to 0.73 indicating a significant contribution of SOA mass to total OM. The simulated SOA/total OM ratios (0.25–0.52) were relatively lower than those in the observed ratios, which are calculated using decoupled SOA and POA with a POC/EC ratio (Sect. 2.4), suggesting that POA is overpredicted and SOA is underpredicted in the CAMx–UNIPAR simulations. For both

observations and simulation, SOA/total OM mass ratios in southern areas, such as Fresno and Bakersfield, were higher than those in northern areas, such as San Jose and Sacramento. The northern area, including the Bay Area is located on the upwind side of the southern area. A strong wind appeared in the northern area during the simulation period (Table S7 and Fig. S4), decreasing the residence time of pollutants, which reduced secondary products of pollutants in this region. The concentration of total OM, POA, and SOA increased from 01/23/2018 and maintained the high

OM mass by early February. During this period, the wind speed was relatively weak (Fig. S3) and the PBL height was suppressed (Fig. S5), which is favorable for the accumulation of pollutants in urban areas.

**3.2 Simulation of individual SOA species and their diurnal pattern**

Fig. 4(a–d) shows the time series of the simulated POA and total SOA with the predicted aerosol phase state (wet and dry) at the four sites. Both POA and SOA mass were noticeably high between 01/26/2018 and 02/13/2018. Fig. 4(e–
h) shows the time series of the simulated SOA species and pie charts to represent the averaged fraction of each SOA species during the period in Fig. 4. The predicted total SOA mass was predominated by alkane SOA (33%–76%) and terpene SOA (18%–60%) across all four sites. The contribution of alkane SOA and terpene SOA to the local SOA mass burden varied according to the location of the sites. For example, the fraction of alkane SOA was higher than that of terpene SOA in all sites except Sacramento. The terpene concentrations can be high at the Sacramento site due
to the proximity to the northern and eastern mountain ranges. Overall, the contributions of the isoprene SOA and aromatic SOA to the total SOA mass were small (~5%) and that of sesquiterpene SOA was nearly negligible.

A high fraction of OMP mass attributed to the total SOA ranging from 0.59 to 0.77 as shown in Fig. 4(e–h). In particular, the alkane OMP fraction of total SOA ranged from 0.2 to 0.41, and the terpene OMP fraction of total SOA ranged from 0.15 to 0.53. In the current UNIPAR model, the low-volatility products produced from the oxidation of
long-chain alkanes can significantly increases SOA growth via their gas-particle partitioning to the pre-existing OM. Additionally, the model includes the low volatility products originating from autoxidation of α-pinene ozonolysis products (Roldin et al., 2019; Crounse et al., 2013; Bianchi et al., 2019). The importance of autoxidation mechanisms on terpene SOA formation was demonstrated in a recent study by Yu et al. (2021c) for daytime chemistry. In their study, the peroxy radical autoxidation mechanism (PRAM) developed by Roldin et al. (2019) was included in
UNIPAR to evaluate the impact of HOMs on terpene SOA formation. In the sensitivity test of SOA prediction associated with PRAM, α-pinene SOA mass increased by 15%–35% in the presence of PRAM, suggesting that substantial impact of PRAM on the total SOA mass (Yu et al., 2021c).

In addition, the chamber study and the UNIPAR simulation by Han and Jang (2023) showed a significant potential for nighttime biogenic SOA formation. In their study, α-pinene and isoprene SOA growth during nighttime is even
more significant than their daytime growth due to the considerable contribution of both ozonolysis of biogenic HCs and the reaction with nitrate radicals, which yield low-volatility products. Ozone is produced via the atmospheric oxidation of HCs integrated with a photochemical cycle of $NO_x$ that extends through nighttime and promotes the formation of a nitrate radical in the presence of $NO_2$. This nitrate radical can attack an alkene in terpenes to form an alkylperoxy radical, leading to the formation of low-volatility peroxide accretion products (ROOR) (Hasan et al., 2021;
Bates et al., 2022; Han and Jang, 2023).

The fractions of OMH in anthropogenic SOA (aromatic and alkane SOA) ranged from 0.41 to 0.47 and those of biogenic SOA ranged from 0.13 to 0.18. The anthropogenic OMH fractions are consistent with reported values predicted by Pye and Pouliot (2012) in regional scales in the US domain. For example, they reported that the oligomeric SOA simulated mainly from alkanes and partly from PAHs accounted for about half of the anthropogenic
SOA in the US domain. In the regional simulation by Yu et al. (2022), the SOA mass were simulated with both the

two-product SOA (SOAP) model and the UNIPAR SOA model during the KORUS-AQ campaign. The fraction of OMH to the total SOA are on average 0.31 and 0.40 with the SOAP SOA model and the UNIPAR SOA model, respectively, (Yu et al., 2022). The alkane OMH fractions of the total SOA ranged from 0.13 to 0.35, which indicates that alkane OMH is significant, although less than alkane OMP (Fig. 4e-h). In the UNIPAR model, this alkane OMH is mainly associated with non-volatile products originating from the autoxidation of long-chain alkanes. Unlike alkanes, other precursor HCs yield OMH via oligomerization of reactive semi-volatile organic species. Under California's 2018 winter simulation, the alkane OMH was larger than that sourced from non-alkane OMH. The attribution of autoxidation-driven low volatility products (alkane and terpene) to the total SOA mass can also promote gas-particle partitioning of other products (non-autoxidation products), by increasing the preexisting OM. Furthermore, OMP is dependent on temperature with a negative relation with temperature (Odum et al., 1996). The contribution of OMH to the total SOA can also be impacted by HC species.

Fig. 5 shows the diurnal variation of SOA species averaged between 01/23/2018 and 02/24/2018. The alkane SOA increased during the daytime because alkanes are mainly oxidized by reactions with an OH radical. As discussed above, terpene can be oxidized with an OH radical, a dominant oxidant in daytime, and furthermore terpene reacts with ozone and nitrate radicals during nighttime to form low volatility products, which can be a significant source of OMP. The time series of the simulated concentrations of $NO_2$, ozone, and nitrate radicals are shown in Fig. S7. The time series of simulated SOA precursor HC concentrations (i.e., aromatic HCs, alkane, terpene, sesquiterpene, and isoprene) are shown in Fig. S8. Fig. S9 and Fig. S10 show the time series of emissions of precursor HCs and SOA species, respectively. The terpene SOA mass at Fresno and Bakersfield noticeably increased (Fig. S10) between 01/29/2018 and 02/11/2018 when the concentrations of ozone and nitrate radicals were high (Fig. S7). During the simulation period, no specific fluctuations appeared in both terpene concentrations (Fig. S8) and terpene emissions (Fig. S9), suggesting that terpene SOA formation is positively influenced by the concentrations of ozone and nitrate radicals.

### 3.3 Spatial distribution of SOA species

The surface-level spatial distributions of the total SOA, POA, and individual SOA species, averaged between 01/23/2018 and 02/24/2018, are displayed in Fig. 6. The spatial distribution of individual SOA species can be influenced by the emission of each HC, available atmospheric oxidants, the reactivity of each HC with atmospheric oxidants, and winds (speed and direction). Fig. 2(c–h) illustrates the spatial distributions of the emission of POA and each HC, and Fig. S11(a–c) shows those of $NO_2$ and atmospheric oxidants (ozone and nitrate radicals), respectively. The spatial distribution of the concentrations of each HC is shown in Fig. S11(d–h).

Biogenic HCs are rapidly consumed day and night, due to their fast rate constants with OH radicals, ozone, and nitrate radicals, and quickly form SOA. Hence, the spatial distributions of biogenic SOA were substantially influenced by the distribution of precursor HC emission sources. As shown in Fig. 6(e), the terpene SOA mass was evenly distributed along the Central Valley, which was influenced by the high terpene emission from the surrounding mountains including the Cascade Range to the north, the Sierra Nevada to the east, and the Tehachapi Mountains to the south. On the other hand, high $NO_2$ concentrations appeared in urban areas of the Central Valley as shown in Fig. S11(a).

Both ozone and nitrate radicals react with biogenic HCs to effectively form SOA via the production of low volatility products as discussed in Sect. 3.2 (Li et al., 2011; Jenkin, 2004; Han and Jang, 2023). The formation of nitrate radicals is favorable in urban areas because of the high concentrations of $NO_2$. Owing to the reaction of terpene with this nitrate radicals, the concentration of nitrate radicals was, however, low in the Central Valley. The maximum terpene SOA concentration appeared between the terpene ridge (Fig. S11f) and the $NO_2$ ridge (Fig. S11a) where the reaction rate between terpene and nitrate radicals was high. The sesquiterpene SOA distribution (Fig. 6f) was slightly southward compared with the terpene SOA distribution. In addition to emissions from mountain ranges, sesquiterpene can be sourced from various areas as shown in Fig. 2(g). California's isoprene emissions (Fig. 2h) are expected to originate from oak trees and from eucalyptus trees and are distributed in the central California foothills and South Coast Ranges. Thus, the isoprene SOA distribution (Fig. 6g) was oriented more southward than that of the terpene SOA. Both sesquiterpene SOA and isoprene SOA concentrations were extremely small as discussed in Sect 3.2.

The spatial distributions of anthropogenic SOA were noticeably impacted by the southward wind direction. For example, the spatial distributions of both aromatic SOA (Fig. 6c) and alkane SOA (Fig. 6d) were extremely southward, although their emissions were relatively even along with the Central Valley (Fig. 2d and Fig. 2e). Unlike biogenic HCs, both aromatic and alkane HCs have a relatively long lifetime for their atmospheric oxidation. Additionally, their major oxidation path with OH radicals occurs in daytime. Thus, anthropogenic hydrocarbons can be transported along with the southward wind during the simulation period, which can increase the formation of anthropogenic SOA in the southern area. A $PM_{2.5}$ particle has an average lifetime of approximately eight days (Hodan and Barnard, 2004; Chowdhury et al., 2022). Noticeably, the POA also drifted to the south due to the transportation by the southward wind (Fig. 2c vs. Fig. 6b) and the high POA concentration augmented anthropogenic SOA growth in the southern area.

**3.4 Impact of aerosol hygroscopicity on SOA formation**

Numerous studies (Lim et al., 2010; Liu et al., 2012; Mcneill, 2015; Srivastava et al., 2022) have revealed that aqueous reactions of hydrophilic, reactive organic species in electrolytic inorganic aerosol produce non-volatile oligomeric matter and increase SOA mass. These heterogeneous reactions are known to be accelerated by acid catalyst species (Jang et al., 2002; Hallquist et al., 2009). In order to simulate the SOA formation potential by aerosol acidity at the four sites, the time series of hourly simulated concentrations of ionic species (sulfate, nitrate, and ammonium in Fig. S12) were applied to estimate the aerosol acidity by the charge balance of the inorganic electrolytes. In Fig. S13, the charge balance estimated using the simulated and observed ionic concentrations suggested that the aerosol acidity was nearly neutral. The one-to-one line shown in Fig. S13 shows the fully titrated inorganic composition. Thus, SOA growth by the acid-catalyzed heterogeneous reactions of reactive lumping species might not have been effective during the simulation period of this study. Overall, nitrate concentrations in aerosol were high in the urban areas of California during the winter of 2018. The aerosols, which are enriched with ammonium nitrate salt, can be hygroscopic and lower the aerosol ERH and DRH (Fig. S6). As described in Sect. 2.6, the inorganic aerosol phase was mostly wet due to the low ERH and DRH with the abundant nitrate ammonium salt.

The sensitivities of different SOA species to aerosol hygroscopicity are dissimilar. For example, aromatic and isoprene oxidation products are reactive for heterogeneous reactions and hydrophilic. Thus, they can effectively yield SOA in

inorganic salted aqueous phase (Yu et al., 2021b). In contrast, both alkane and terpene oxidation products are relatively hydrophobic. Thus, their solubility in the aqueous phase is low due to their large activity coefficients. Evidently, α-pinene SOA and alkane SOA have a low O:C ratio (~0.55) (Zhang et al., 2015; Galeazzo et al., 2021; Madhu et al., 2022) compared with isoprene SOA (~1.1) (Song et al., 2015). In order to evaluate the sensitivity of different SOA species to aerosol hygroscopicity, the isoprene to alkane SOA mass ratios were calculated during the wet and dry periods as shown in Fig. 7. The SOA mass ratios of isoprene to alkane during the wet period were higher than those during the dry periods for all sites: San Jose, Sacramento, Fresno, and Bakersfield by factors of 3.0, 1.9, 2.8, and 2.0, respectively. Such high SOA mass ratios of isoprene to alkane during the wet period indicate the importance of the role of aerosol hygroscopicity on the SOA formation in an environment with a high emission of isoprene or aromatic HCs. The impact of aerosol hygroscopicity on SOA mass was minimal under California's atmospheric environment during the simulation period because of the dominance of alkane and terpene SOA. In contrast, the SOA mass simulated during the KORUS-AQ campaign was dominated by aromatic HCs, which produced reactive and hydrophilic products (Yu et al., 2022). In their study, the CAMx–UNIPAR model well captured a significant contribution of aromatic SOA to the total OM during the low-level transport/haze period (Crawford et al., 2021) where the aerosol water content was high.

**3.5 Impact of NO$_2$ on SOA formation**

To understand the impact of NO$_2$ on SOA formation, the correlation coefficients (R) between NO$_2$, nitrate aerosol, and SOA mass associated with each HC were estimated as shown in Table 3. The analysis of R between NO$_2$, nitrate and SOA was performed during high PBL height periods (02/11/2018–02/24/2018).  The formaldehyde-to-NO$_2$ ratio (FNR) is typically used to indicate NO$_x$ levels. When FNR is less than 1, it represents VOC-limited condition (Duncan et al., 2010; Hoque et al., 2022). Based on the spatial distribution of FNR in Fig. S14, the four urban sites of this study were VOC-limited (high NO$_x$ levels) conditions. Under this environment, a typical SOA production in daytime is negatively correlated with NO$_2$ concentrations (Presto et al., 2005; Yang et al., 2020; Madhu et al., 2022). However, all correlation coefficients between NO$_2$ and SOA mass shown in Table 3 were positive. Biogenic SOA was more strongly correlated with NO$_2$ (larger positive R) than anthropogenic SOA.

As discussed in Sect. 3.2 and Sect. 3.3, biogenic HCs, particularly terpene, react with a nitrate radical to form low-volatile products and effectively increase SOA mass. NO$_2$ is linked to the formation of nitrate radicals at nighttime, and it can be positively related to biogenic SOA mass.  Increased nighttime biogenic SOA mass (Fig. 5) can also influence OMP of anthropogenic SOA, weakening the negative correlation between NO$_2$ and anthropogenic SOA formation. For example, the last column in Fig. 5, which displays the diurnal variation of the SOA concentrations, evinces the influence of terpene SOA on anthropogenic SOA mass. Anthropogenic SOA mass still gradually increased after sunset (5 PM) when the production of OH radical was nearly inactive, and it continued by midnight (i.e., 12AM at Fresno). As shown in Fig. S15, a rapid change in the PBL height appeared between 3PM and 5PM, and no change appeared in the PBL height after 7PM. Thus, the influence of PBL height changes on anthropogenic SOA mass after 7PM can be trivial.

Under the Central Valley's environments in this study, ammonia was rich, temperature was relatively low, and humidity was high. The high concentration of $NO_2$ in this region is favorable to form inorganic nitrate aerosol, which can promote aqueous phase reactions of organic species. Evidently, a strong positive correlation appeared between inorganic nitrate concentrations and SOA mass (R: 0.41–0.86 in Table 3) declining a conventional negative correlation between $NO_x$ concentration and anthropogenic SOA formation at high $NO_x$ levels.

**4 Atmospheric Implications and uncertainties**

The CAMx–UNIPAR model utilized 251 lumping species, which were constructed by using the semi-explicitly predicted products for three major oxidation paths (OH radicals, ozone, and nitrate radicals) of a wide range of HCs including aromatics, long-chain alkanes with different carbon lengths, isoprene, terpene, and sesquiterpene. Semi-explicit gas mechanisms captured the emerging chemistry in both the gas and aerosol phases as a function of the $NO_x$ level and gas aging. In CAMx–UNIPAR, the calculation of the activity coefficient of lumping species in the multiphase allowed the simulation of the impact of temperature and humidity on the multiphase partitioning of lumping species and their heterogeneous reactions in the aerosol phase to form oligomeric OM. The OM budget was reasonably simulated by CAMx–UNIPAR for the urban environments in the Central Valley during the winter of 2018 (Fig. 3). However, the predicted POA was slightly overpredicted and the predicted SOA was somewhat underestimated compared with the decoupled POA and SOA from the observed OC.

In-depth analyses of individual SOA species in the Central valley under California's environment showed that the total SOA mass was predominated by daytime oxidation of long-chain alkanes ranging from C9 to C24 and nighttime terpene oxidation with ozone and nitrate radicals (Sect. 3.2). SOA growth considerably increased with the low volatility products originating from the autooxidation of terpene and alkane oxidation products, and the nighttime chemistry of biogenic HCs. The prevalence of alkane and terpene SOA was the unique feature of California although their relative fractions to total OM may be switched seasonally. Unlike California, a significant SOA species in East Asia was aromatic SOA because of the high emissions of aromatic HCs as seen in the CAMx–UNIPAR simulation during the KORUS-AQ campaign (Yu et al., 2022). We also concluded that the significance of inorganic hygroscopicity on SOA formation is influenced by the major SOA species. The aerosol's hygroscopicity effect on SOA formation is particularly pronounced in the region rich in isoprene or aromatic HC. Seemingly, the sensitivity of SOA formation to inorganic aerosol hygroscopicity in California was somewhat weaker than that simulated in South Korea during the KROUS-AQ campaign. In the future, the CAMx–UNIPAR model needs to be demonstrated to predict SOA burdens in the Southeastern US, where isoprene is abundant due to its high emission from deciduous tree species (Palmer et al., 2003; Mauzerall et al., 2005) and it is humid, especially in summer.

An important feature of this study was the characterization of the influence of HC's atmospheric lifetime on the spatial distributions of each SOA species (Fig. 6). The simulated terpene SOA appeared in the region proximate to terpene emission sources because of terpene's rapid oxidation day and night. Unlike terpene, the spatial distributions of aromatic and alkane HCs, which are relatively slowly oxidized in daytime with an OH radical, were influenced by the wind direction. Both alkane SOA and aromatic SOA were high in the south of the Central Valley due to the southward wind during wintertime. The similar tendency was simulated during the KORUS-AQ campaign (Yu et al., 2022).

Aromatic HC emissions are high in the industrial areas of Shandong Province in East China, and aromatic SOA was eastward along the wind direction. However, terpene SOA was almost synchronized with terpene sources, which were high from southeastern mountains in China.

Due to efforts by the government, $NO_x$ and $SO_2$ emissions from anthropogenic sources have been gradually reduced (Winkler et al., 2018; Zheng et al., 2018; Yu et al., 2021a; Aas et al., 2019). The sensitivity of each SOA species to the $NO_x$ level can be various according to their oxidation mechanisms. For example, chamber studies showed that aromatic SOA yields increased with decreasing the $NO_x$ level within the $NO_x$-rich region (Yang et al., 2020). A gradual increase in alkane SOA was reported in the recent chamber study by Madhu et al. (2022) with decreasing the $NO_x$ level over both $NO_x$-rich and $NO_x$-poor regions. The reduction of $NO_x$ in the $NO_x$-rich region dramatically decreases nighttime biogenic SOA yields owing to the reaction of biogenic HCs with decreased nitrate radicals, but the reduced $NO_x$ increases daytime biogenic SOA yields (Han and Jang, 2023). As discussed in Sect. 3.5, the anthropogenic SOA formation can be influenced by the amount of biogenic SOA via gas-organic phase partitioning of organic products, and vice versa. Hence the impact of $NO_2$ on SOA formation can be varied with the composition of precursor HCs. The SOA mass in California urban areas possibly would decrease as $NO_x$ decreases because of the dominance of terpene SOA, which has a strongly positive correlation with $NO_2$ as shown in Table 3. For the polluted urban area where anthropogenic SOA is dominant and $NO_x$ is rich, SOA formation can be negatively correlated with $NO_x$.

The impact of $SO_2$ changes on SOA formation is complicated due to the aerosol acidity and the amount of wet-inorganic aerosol. The reduction of $SO_2$ reduces the SOA formation via oligomerization of organics, which is catalyzed by aerosol acid. However, the reduced aerosol acidity with decreasing $SO_2$ under ammonia-rich environments can increase the deposition of nitric acid forming very hygroscopic ammonium nitrate. Both ERH and DRH drop as increasing nitrate fraction in ammonium-sulfate-nitrate aerosol. Reduced ERH and DRH increases wettability of aerosol. Elevated ammonium nitrate mass increases partitioning of polar organics onto the wet aerosol and enhances oligomerization of reactive organic species in aqueous phase.

The possible uncertainties in the SOA simulation using the CAMx–UNIPAR model are associated with missing precursors, the classification of precursors due to uncertainties in emission data and the use of surrogates, and the mass balance between gas chemistry and the aerosol module. Numerous precursors are still missing in UNIPAR because of the lack of emission data and model parameters of HCs such as phenols, branched and cyclic alkane, and polyaromatic hydrocarbons. In order to reduce the computational time, some HC classes use a surrogate compound. For example, α-pinene was a surrogate for all terpenes and this substitution causes uncertainties in terpene SOA due to the different reactivity to oxidants, the formation of autoxidation products and SOA yields (Yu et al., 2021c). In this study, all alkanes were surrogated by linear alkanes. Overall, branched alkanes have lower SOA yields than linear alkanes at a given carbon number, whereas cyclic alkanes have higher SOA yields. Additionally, emissions of long-chain alkanes might be uncertain. The deposition flux of IVOCs, such as long-alkanes, onto PM and various ground-level surfaces can increase with increasing IVOC molecular weight due to their low volatility. Hitherto, the impact of partitioning on IVOCs emissions or the dry deposition is poorly treated in the regional models. Omitting dry deposition of IVOCs could overpredict SOA production (Bessagnet et al., 2010).

Pye et al. (2022) recently developed the Community Regional Atmospheric Chemistry Multiphase Mechanism (CRACMM v1.0), which employs explicit oxidized products to improve the carbon balance between gas mechanisms and the SOA model (Pye et al., 2022). In CAMx–UNIPAR, the carbon balance between gas chemistry and the aerosol module, which is caused by consumption of carbon to form SOA, has not been addressed. Interconnecting atmospheric gas-chemistry with SOA formation is essential because SOA formation is a dynamic process as a function of

precursors HCs, $NO_x$, $SO_2$, and atmospheric oxidants. In order to expand the suitability of CAMx–UNIPAR to more field observations, emission species of precursors HCs need to be integrated with Carbon Bond 6 mechanism (CB6) (Yarwood et al., 2010), which is frequently used for HC gas oxidation.

Code availability. Code to run the CAMx–UNIPAR model in this study is available upon request.

Data availability. Regional simulation input data is available upon request.

Author contributions. Y. Jo simulated the air quality by using the CAMx–UNIPAR model for the urban environments in the Central Valley, CA. Y. Jo and MJ interpreted the air quality simulation results. Y. Jo, MJ and ZY integrated

the UNIPAR code and model parameters into CAMx. SH and AM produced model parameters for UNIPAR. BK and Y. Jia provided emission and meteorology inputs for California's air quality simulation and reviewed the paper with comments. SK and JP generated emission split factors for SOA precursors.

Competing interest. The authors declare that they have no conflict of interest.

**Acknowledgments**

This research was supported by the National Institute of Environmental Research (NIER2020-01-01-010); the National Science Foundation (AGS1923651); and the National Research Foundation of Korea funded by the Ministry of Science and ICT (2020M3G1A1114556). Emissions and meteorological inputs for the 2018 California regional modeling were provided by the Bay Area Air Quality Management District.

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

**Table 1. List of SOA precursors and SOA species in the UNIPAR model.**

| Type of precursors | UNIPAR species name (OMH, OMP)[a] | Hydrocarbons | Stoichiometric coefficient ($\alpha_i$)[b] | Unified physico-chemical parameters ($MW_i$, $O{:}C_i$, and $HB_i$) | Chamber data[c] | Regional simulation |
|---|---|---|---|---|---|---|
| Aromatics | OMH_AR OMP_AR | Benzene | 50×4 | 50 | (Im et al., 2014; Zhou et al., 2019) | (Yu et al., 2022) |
| | | Toluene | 50×4 | | | |
| | | Ethylbenzene | 50×4 | | | |
| | | Propylbenzene | 50×4 | | | |
| | | *o*-Xylene | 50×4 | | | |
| | | *m*-Xylene | 50×4 | | | |
| | | *p*-Xylene | 50×4 | | | |
| | | 1,2,3-trimethylbenzene | 50×4 | | | |
| | | 1,2,4-trimethylbenzene | 50×4 | | | |
| | | 1,3,5-trimethylbenzene | 50×4 | | | |
| Alkanes | OMH_AK OMP_AK | Alkane (C9–C14) | 50×4 | 50 | (Madhu et al., 2022) | This study |
| | | Alkane (C15–C19) | 50×4 | | | |
| | | Alkane (C20–C24) | 50×4 | | | |
| Terpene | OMH_TE OMP_TE | Terpene with OH radicals | 50×4 | 50 | (Yu et al., 2021c; Han and Jang, 2023) | This study |
| | | Terpene with ozone | 50×4 | | | |
| | | Terpene with nitrate radicals | 50×4 | | (Han and Jang, 2023) | |
| Sesquiterpene | OMH_SP OMP_SP | Sesquiterpene with OH radicals | 50×4 | 50 | (Han and Jang, 2023) | This study |
| | | Sesquiterpene with ozone | 50×4 | | | |
| | | Sesquiterpene with nitrate radicals | 50×4 | | | |
| Isoprene | OMH_IS OMP_IS | Isoprene with OH radicals | 51×4 | 51 | (Beardsley and Jang, 2016; Han and Jang, 2023) | This study |
| | | Isoprene with ozone | 51×4 | | | |
| | | Isoprene with nitrate radicals | 51×4 | | (Han and Jang, 2023) | |

[a] OMH is the SOA mass heterogeneously formed and OMP is that produced via multiphase partitioning.

[b] Four different mathematical equation sets for $\alpha_i$. For aromatics and alkanes, the mathematical equation sets for $\alpha_i$ include 'Low $NO_x$-Fresh', 'Low $NO_x$-Aged', 'High $NO_x$-Fresh', and 'High $NO_x$-Aged'. For terpene, sesquiterpene, and isoprene, the mathematical equation sets for $\alpha_i$ include 'Low $NO_x$-Daytime', 'Low $NO_x$-Nighttime', 'High $NO_x$-Daytime', and 'High $NO_x$-Nighttime'.

[c] Demonstration of SOA model parameters by using data from UF-APHOR chamber studies.


**Table 2. List of observation sites.**

|  | San Jose | Sacramento | Fresno | Bakersfield |
|---|---|---|---|---|
| Latitude | 37.35 | 38.61 | 36.79 | 35.36 |
| Longitude | -121.89 | -121.37 | -119.77 | -119.06 |
| Elevation (above MSL) | 31m | 8m | 96m | 0m |
| Local Type | Urban and Center City | Suburban | Urban and Center City | Urban and Center City |
| Geography | The San Francisco Bay Area Air Basin (SFBAAB or Bay Area) |  | The Central Valley |  |


**Table 3. Correlation coefficients (R) between NO$_2$, nitrate (NO$_3^-$), and SOA species mass at four sites. NO$_2$, nitrate and SOA species concentrations during the high PBL height period (02/11/2018-02/24/2018)[a] were used to minimize the impact of pollutant accumulation due to the suppressed vertical diffusion.**

| | Correlation coefficient (R) | | | |
|---|---|---|---|---|
| | **San Jose** | **Sacramento** | **Fresno** | **Bakersfield** |
| NO$_2$—Nitrate (NO$_3^-$) | 0.66 | 0.68 | 0.53 | 0.51 |
| NO$_2$—Total SOA | 0.72 | 0.68 | 0.61 | 0.66 |
| NO$_2$—Aromatic SOA | 0.42 | 0.60 | 0.40 | 0.44 |
| NO$_2$—Alkane SOA | 0.49 | 0.63 | 0.48 | 0.56 |
| NO$_2$—Terpene SOA | 0.84 | 0.63 | 0.73 | 0.82 |
| NO$_2$—Sesquiterpene SOA | 0.70 | 0.64 | 0.60 | 0.53 |
| NO$_2$—Isoprene SOA | 0.64 | 0.51 | 0.52 | 0.48 |
| Nitrate—Total SOA | 0.86 | 0.68 | 0.86 | 0.85 |
| Nitrate—Aromatic SOA | 0.69 | 0.73 | 0.79 | 0.83 |
| Nitrate—Alkane SOA | 0.81 | 0.73 | 0.82 | 0.84 |
| Nitrate—Terpene SOA | 0.77 | 0.57 | 0.82 | 0.75 |
| Nitrate—Sesquiterpene SOA | 0.62 | 0.41 | 0.73 | 0.65 |
| Nitrate—Isoprene SOA | 0.63 | 0.63 | 0.82 | 0.70 |

[a] Refer to Fig. S5


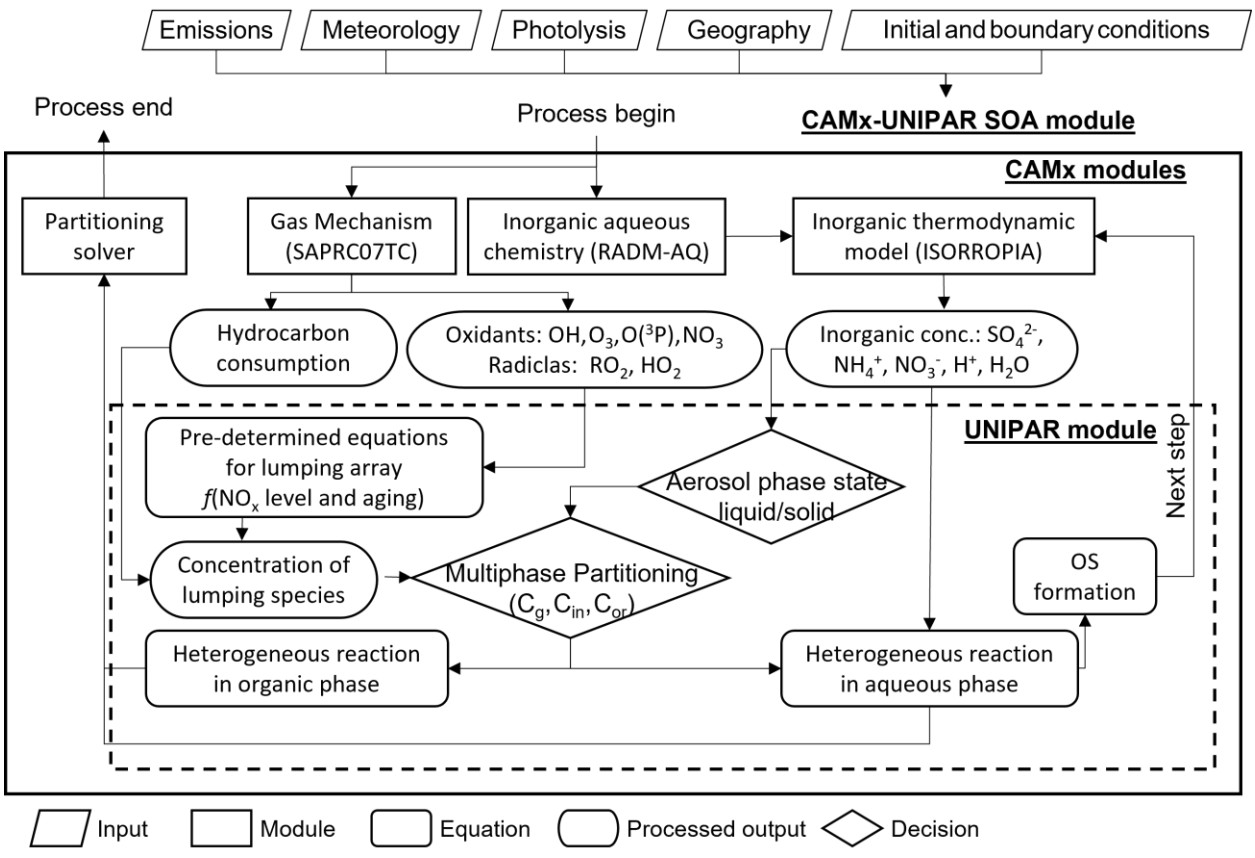

**Fig. 1. The structure of the CAMx model integrated with UNIPAR to form SOA via multiphase reactions of hydrocarbons.**

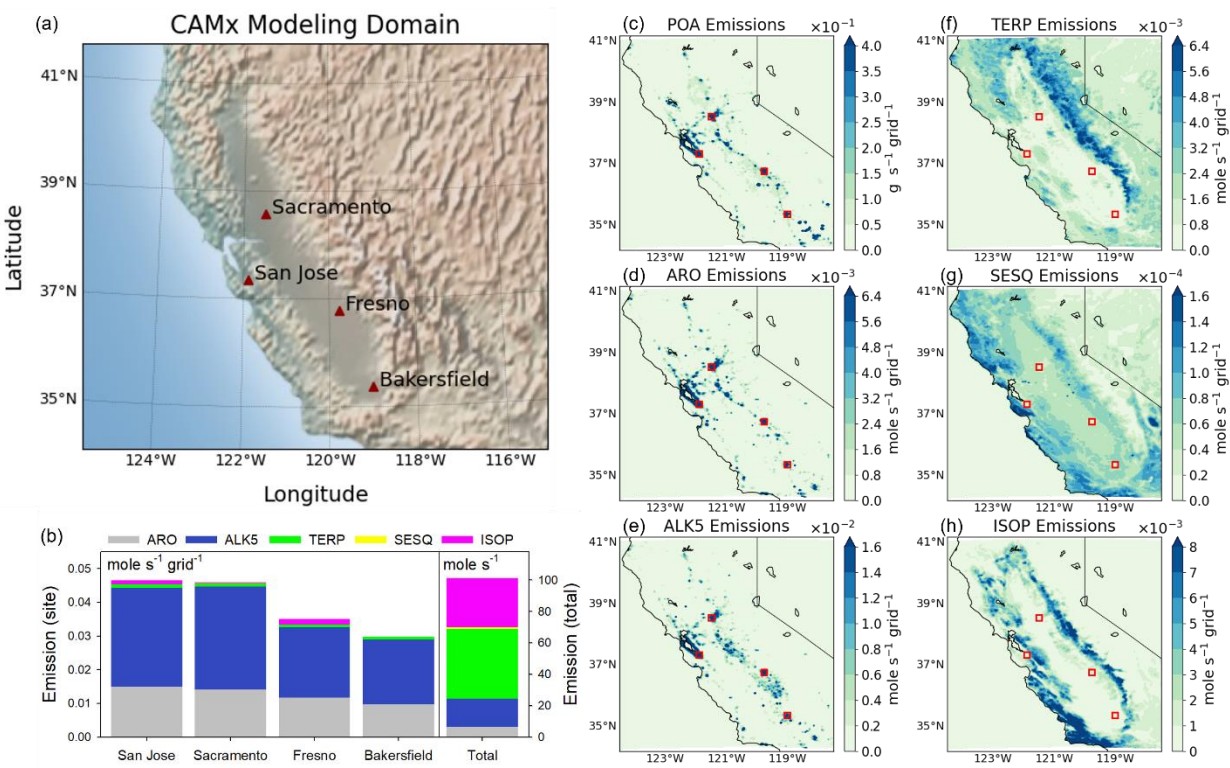

Fig. 2. (a) The CAMx modeling domain and selected observation sites in this study. (b) The amount of major SOA precursors emission for each site and the total of the domain. (c–h) Spatial distributions of POA and major SOA precursors emission. The emission data is SAPRC07-based and averaged from January to February 2018. ARO is the sum of benzene, toluene, $o$-xylene, $p$-xylene, $m$-xylene, and 1,2,4-trimethylbenzene.

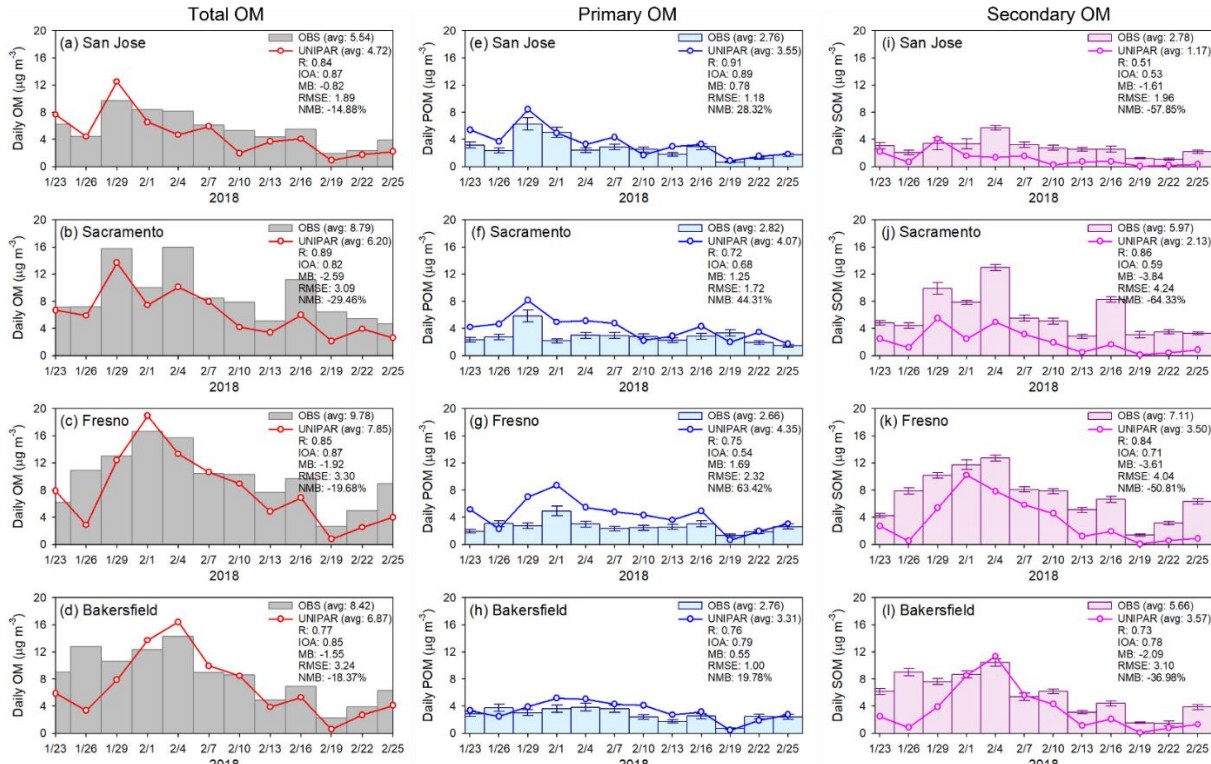


**Fig. 3. (a–d)** Time series of observed and simulated total OM mass concentration (μg m⁻³) at the CSN (Chemical Speciation Network) sites selected in this study. **(e–h)** Time series of observed POA and simulated POA. **(i–l)** Time series of observed SOA and simulated SOA. Vertical bars represent observed POA and SOA estimated using a value of 2.4 for the POC/EC ratio, while the error bars represent a range of POC/EC ratio from 2.05 to 2.75. The OM mass concentrations are displayed in a daily averaged value every three days since the CSN sites operate on a 1-in-3-day sample collection schedule. The equations for statistical parameters are described in Table S8. R is correlation coefficient, IOA is Index of Agreement, MB is Mean Bias, RMSE is Root Mean Square Error, and NMB is Normalized Mean Bias.

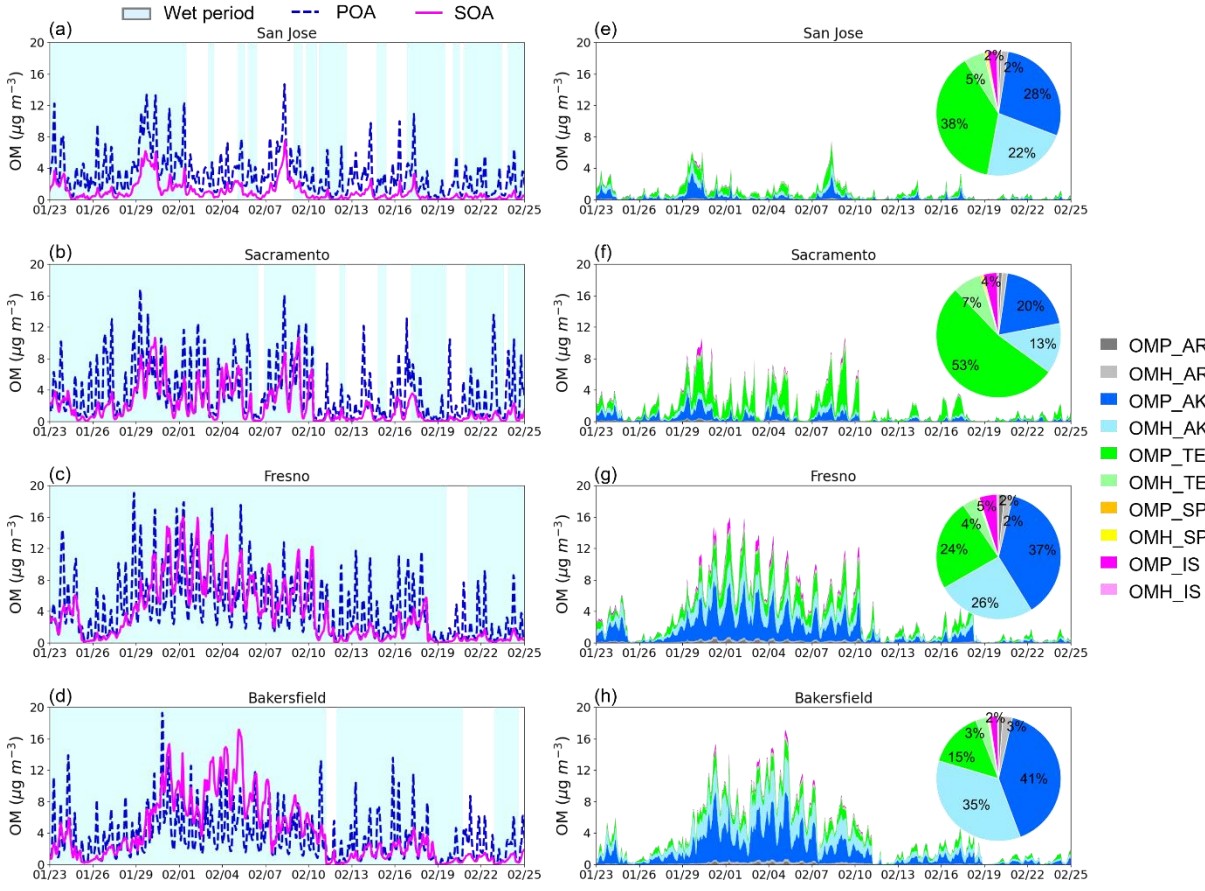

Fig. 4. (a–d) Time series of simulated POA, SOA, and salted aerosol phase state (wet and dry). (e–h) Time series of simulated SOA species in the UNIPAR model at each site and the pie charts to represent the fraction of SOA species associated with graphs (e–h).

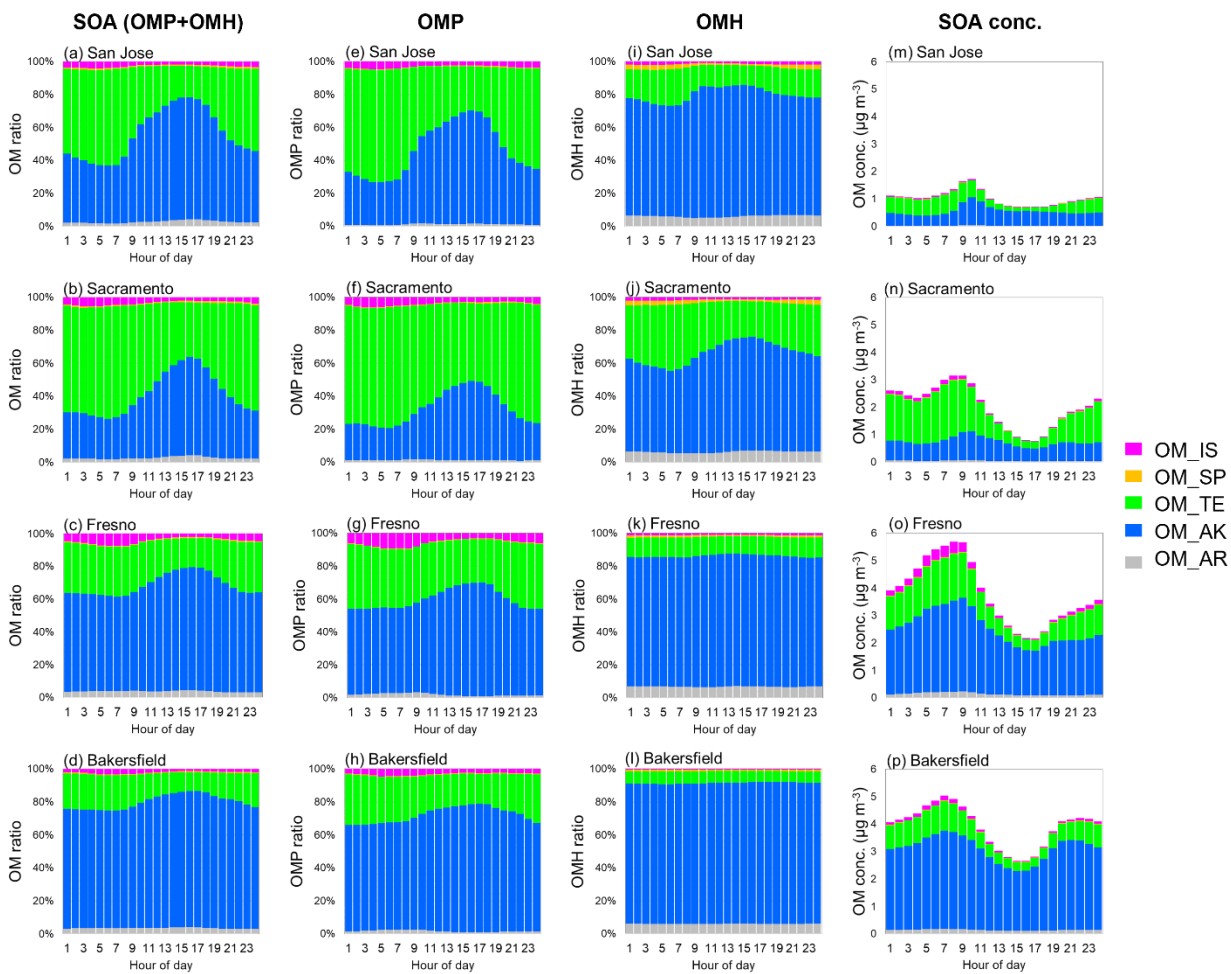

**Fig. 5.** Averaged diurnal variations of (a–d) the total SOA (OMP + OMH) species (%), (e–h) the OMP SOA species (%), (i–l) the OMH SOA species (%), and (m-p) the total SOA (OMP + OMH) species (μg m$^{-3}$) during the simulation period (01/23/2018-02/24/2018).

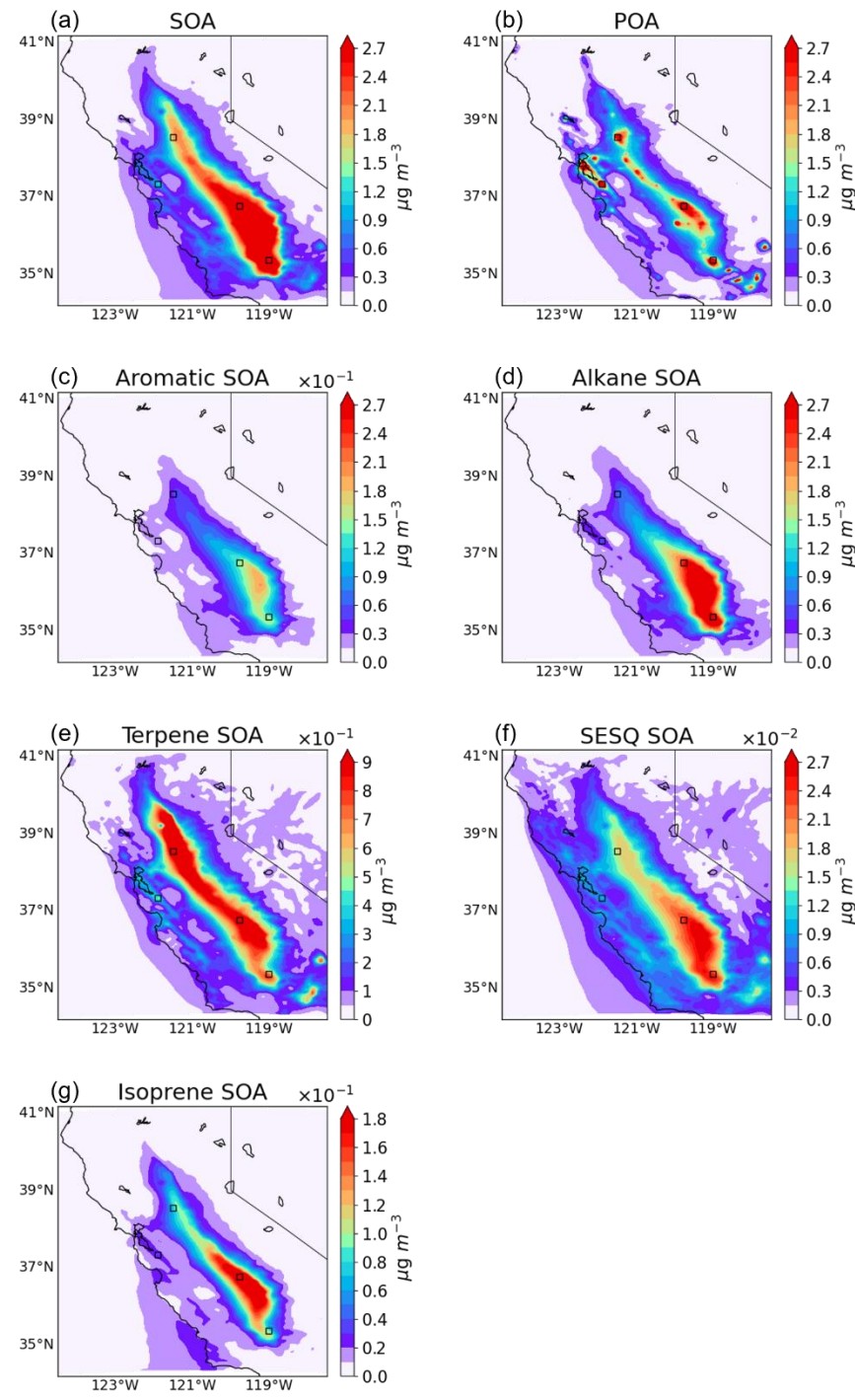

Fig. 6. Spatial distribution of (a) total SOA, (b) POA, (c) aromatic SOA (OMP_AR+OMH_AR), (d) alkane SOA (OMP_AK+OMH_AK), (e) terpene SOA (OMP_TE+OMH_TE), (f) sesquiterpene SOA (OMP_SP+OMH_SP), and (g) isoprene SOA (OMP_IS+OMH_IS). The simulation results at the surface level were averaged between 01/23/2018 and 02/24/2018.

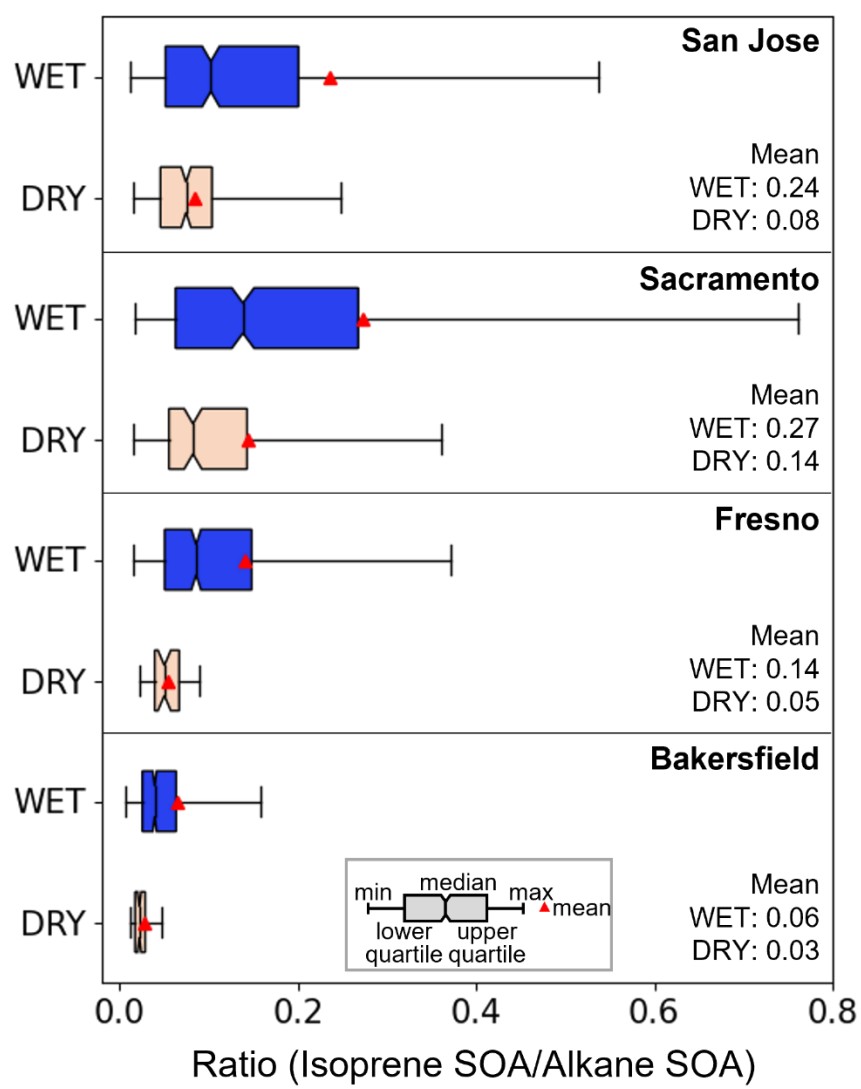

930

**Fig. 7. The mass ratio of isoprene SOA to alkane SOA at the San Jose, Fresno, Sacramento, and Bakersfield during the wet and dry periods.**