# Peer review of "CAMx-UNIPAR Simulation of SOA Mass Formed from Multiphase Reactions of Hydrocarbons under the Central Valley Urban Atmospheres of California"

_EGUsphere, 2023_

## Author Comment (AC1)

**Manuscript #: egusphere-2023-93**

**Response to Anonymous Referee #1**

We would like to thank the reviewer for the time and the constructive comments on our work. The comments are reproduced below along with the author response. The changes made to the manuscript or supporting information were in red color.

*Comment 1*

*The authors should better state how innovative their work is. Except the Yu et al., 2022 paper for the Korus campaign in South Korea, are their other examples where a detailed multiphase aerosol model like UNIPAR was integrated into a 3D model system? This should be pointed out.*

**Response:** Very rarely have detailed multiphase aerosol models, such as UNIPAR, been applied to the 3D model. We added the novelty of UNIPAR in Section 2.1 and reads now,

"The CAMx-UNIPAR of this study simulates SOA formation via multiphase reactions of ten aromatics, three biogenics (isoprene, terpene, and sesquiterpene), and three alkane groups as summarized in Table 1. The lumping species, which are generated by using semi-explicitly predicted products of the oxidation of HCs, are involved in multiphase partitioning and aerosol-phase reactions to form SOA mass. The prediction of stoichiometric coefficients associated with the lumping species from the oxidation of ten aromatics (MCM v3.3.1(Jenkin, 2004)) potentially increases the accuracy of the prediction of aromatic SOA formation under varying emissions. For biogenic HCs, recently identified biogenic oxidation mechanisms that yielded low-volatility products have been applied to the prediction of lumping species. For example, the peroxy radical autoxidation mechanism (Roldin et al., 2019) is known to form highly oxygenated organic molecules (HOMs) (Molteni et al., 2019). Additionally, HOMs form via the accretion reaction to form ROOR from $RO_2$ species (Bates et al., 2022; Zhao et al., 2021). These HOMs are included in biogenic lumping species in the current CAMx-UNIPAR. In order to represent day and night chemistry, CAMx-UNIPAR is equipped with the oxidation-path-dependent stoichiometric coefficients from individually processed reactions of biogenic HCs with three major oxidants (OH radicals, ozone, and nitrate radicals) (Han and Jang, 2023). Thus, nighttime SOA formation, which is dominated by oxidation via ozone and nitrate radicals, can be simulated in CAMx-UNIPAR. By using the recent chamber study by Madhu et al. (2022), the lumping species generated from the oxidation of long-chain alkanes raging from C9 to C24 have been newly integrated with CAMx-UNIPAR, which increases the predictability of SOA formation from intermediate volatility organic compounds (IVOCs). Alkane gas mechanisms include autoxidation of alkoxy radicals to form HOMs (Madhu et al., 2022; Crounse et al., 2013; Bianchi et al., 2019; Roldin et al., 2019). Importantly, the estimation of activity coefficients of lumping species in both the organic phase and the inorganic aqueous-phase enables the simulation of multiphase partitioning of lumping species and their aerosol phase reactions. Aqueous reactions in CAM-UNIPAR facilitate the evaluation of the impact of humidity, aerosol acidity, and aerosol-phase state on SOA formation in regional scales."

*Comment 2*

*While the Central valley might be an interesting place with significant organic aerosol pollution, the available data from four sites of the standard air quality network are in my sense rather poor for detailed model evaluation. They only provide daily averages each three days, so we have 12 OC data points for each of the four sites. I would propose two options: either apply the model to an intensive field campaign giving information on precursor gases, oxidants, and organic aerosol composition, ideally including tracers for different formation pathways and precursors. This would be in my mind highly interesting, and I recommend thinking in this way.*

**Response:** The simulation of the air quality in the Central valley, CA was the first case to demonstrate CAMx-UNIPAR to regional simulation in the USA domain. We utilized the emissions of hydrocarbons based on SAPRAC07 gas mechanisms in this simulation. SAPRAC07 was chosen because it can explicitly cover hydrocarbon species. However, preexisting emissions of hydrocarbons was mostly established on CB6 in regional models. The emission based on SAPRAC07 species was available in the database from BAAQMD in CA. BAAQMD provided emission and meteorological input data for the CAMx-UNIPAR simulation. In the future, CAMx-UNIPAR will be tested using different gas mechanism platform such as CB6 to demonstrate the model on various field observations. In order to response to the reviewer, we added the sentence into the end of manuscript and reads now,

"In the future, emission species will be integrated with Carbon Bond 6 (CB6) gas mechanism (Yarwood et al., 2010), which is frequently used for hydrocarbon gas oxidation, and applied to CAMx-UNIPAR to demonstrate the model on various field observations."

**Comment 3**

*As an alternative, authors could conduct a more operational evaluation. To my knowledge, the CSN and IMPOROVE networks contain more the 100 sites with OM data. Performing a simulation over a wider area and period would allow a truly statistical analysis of model ability to simulate SOA. From a computer point of view, is a longer situation possible?*

**Response:** We agree with the reviewer. In the future, CAMx-UNIPAR will be evaluated in different regions and different time. As discussed in the response to comment 2, The CAMx-UNIPAR simulation of this study was restricted by the emission data, which was coupled with hydrocarbon species in SAPRAC07 mechanism. The emission data which have a fixed domain were provided by BAAQMD, so the modeling domain cannot be extended. Furthermore, wildfire occurred frequently in California. We selected the season where wildfire is inactive in 2018, which was winter months.

**Comment 4**

*I think that figures 4 and 5 are not coherent with figure 6. From what I see, figure 6 shows a bit less than half of aromatic SOA with respect to alkane SOA averaged over the period (roughly both contributions have similar spatial contributions) while in figures 4 and 5 the aromatic contribution is very low. The same incoherence seems to appear for sesquiterpene SOA, in figure 6 it is with terpene the major SOA contributor in San José, but in figures 4 and 5 it is nearly absent. In section 2.5 it is said, that sesquiterpene emissions are negligible. Please check these figures.*

**Response:** In Fig. 6, please notice that each SOA species has different scale. To clearly show the color-bar scales, the font size increased in the revised manuscript.

**Comment 5**

*Beyond this incoherence, it is astonishing that the aromatic contribution is so low, and the alkane one so high. In the conclusion it is said that these are partly long alkane chains, probably in the IVOC range. This should be made clearer already before the conclusion, when talking about emissions in section 2.5*

**Response:** The very low contribution of aromatic SOA is strongly associated with the low level of aromatic emissions in CA as shown in Fig. 2. Long chain alkanes significantly contribute to IVOCs, which emitted from automobile exhaust and plants. In this paper, the SOA mass from alkanes were simulated based on emission recently reported by the EPA (Pye et al., 2022) and the study by Mcdonald et al. (2018). Stoichiometric coefficients associated with lumping species were estimated with the recent study by Madhu et al., (2022). Based on this protocol, alkane SOA has been first time simulated using CAMx-UNIPAR. As shown in Fig. 2, alkane is dominant among anthropogenic precursors. In our model, alkane precursors are split into three groups based on their volatility. In particular, the fractions of alkanes which have more than C13 (IVOC) is significant (approximately 60% of all alkanes). The information about alkane has been updated in Section 2.2, and the uncertainty of alkane SOA overprediction has been added in Section 4 as follows,

"Long chain alkanes are important IVOCs, which are emitted from automobile exhaust (Pye and Pouliot, 2012; Ensberg et al., 2014) and plants (Simoneit, 2002; Yao et al., 2009; Li et al., 2022). Laboratory studies showed that SOA yields from long chain alkanes are high suggesting their significance in ambient SOA mass (Lim and Ziemann, 2009; Tkacik et al., 2012; Presto et al., 2010; Srivastava et al., 2022). In the model, alkanes in different carbon chain lengths are split into three groups due to their different SOA formation potential: C9-C14, C15-C19, and C20-C24 (Table 1)." in Section 2.2

"Additionally, the uncertainty of emissions in long-chain alkanes might be high. The deposition flux of IVOCs onto PM and various ground-level surfaces can increase with increasing IVOC molecular weight due to their low volatility. Hitherto, the impact of multiphase partitioning on IVOCs emissions or the dry deposition is poorly treated in the regional models. Omitting dry deposition of IVOCs could overpredict SOA production (Bessagnet et al., 2010)." in Section 4

**Comment 6**

*The detailed model results about species contribution cannot be evaluated from the masurements available from this paper. They should be discussed in depth based on intensive campaigns elsewhere, beyond the discussion with the KORUS campaign. In the same order of ideas, is the 10 -35% simulated contribution of in-particle*

*formation expected from literature, has such a contribution already been simulated, observed in the field or in a chamber?*

**Response:** The contributions of SOA mass in-particle phase (OMH) in this study are generally consistent with the literature. We have added additional discussion about OMH in the revised manuscript (Section 3.2) as below,

"The predicted contributions of OMH to the total SOA using UNIPAR ranged from 23% to 41%. The fractions of oligomeric matter (OMH) from anthropogenic HCs (aromatics and alkanes) ranged from 41% to 47% and those of biogenic HCs ranged from 13% to 18%. The anthropogenic OMH fractions are consistent with reported values predicted by Pye and Pouliot (2012) in regional scales in the US domain. For example, they reported that the oligomeric SOA simulated mainly from alkanes and partly from polycyclic aromatic hydrocarbons (PAHs) accounted for about half of the anthropogenic SOA in the US domain. In the regional simulation by Yu et al. (2022), the SOA mass were simulated with both the two-product SOA (SOAP) model and the UNIPAR SOA model during the KORUS-AQ campaign. The contributions of OMH to the total SOA are on average 31% and 40% with SOAP SOA model and the UNIPAR SOA model, respectively."

*Comment 7*
*The paper's quality in English wording is not sufficient. This can easily be improved. I put some examples below.*
**Response:** The manuscript has been thoroughly reviewed by the native English speaker.

*Comment 8*
*Line 55: "To compensate the underestimation of SOA, the SOA model employed high partitioning-base model parameters, emerging gas mechanisms to form low-volatile products via autoxidation (Mayorga et al., 2022; Jokinen et al., 2015; Pye et al., 2019), or nighttime oxidations of HC with nitrate radical and ozone (Zaveri et al., 2020; Gao et al., 2019)." This suggests that the mentioned processes do not occur in the atmosphere, or at least are overestimated. Both statements are not proven.*
**Response:** The sentence has been revised as below,

"To improve the prediction of SOA, SOA models have employed large partitioning-based parameters, gas mechanisms to form low-volatile products via autoxidation…"

*Comment 9*
*Line 82: "In this study, the CAMx–UNIPAR model was extended to alkane SOA and nighttime chemistry of biogenic HCs."    and Line 87: "In addition, the UNIPAR model has recently been expanded to simulate biogenic SOA based on three major paths (i.e., OH radicals, ozone and nitrate radicals) being capable of nighttime SOA formation (Han and Jang, 2023) that was dominated by oxidation with ozone and nitrate radicals."   From this it is not clear where these changes are first presented, in the present work or already in Han and Jang (2023).   Or the first comparison to field data is done here?   (it becomes clear from table 1 and looking into the papers but should be made clear in the text).*
**Response:** To clearly show the new additions in the CAMx-UNIPAR of this study, the last paragraph of Section 1 has been revised as follows,

"In this study, the CAMx–UNIPAR model has been updated to include the SOA formation from long-chain alkanes and nighttime chemistry of biogenic HCs. Long-chain alkanes are regarded as essential precursors for SOA formation (Aumont et al., 2012; Madhu et al., 2022). Madhu et al. (2022) have recently added an autoxidation mechanism into alkane semi-explicit oxidation mechanisms, improving the predictability of alkane SOA using the UNIPAR model against their chamber study. The resulting alkane model parameters have been newly implemented into CAMx-UNIPAR. In addition, the UNIPAR model of this study has been expanded to simulate biogenic SOA based on three major oxidation paths (i.e., OH radicals, ozone and nitrate radicals) being capable of nighttime SOA formation that is dominated by oxidation with ozone and nitrate radicals (Han and Jang, 2023)."

*Comment 10*
*Around line 190 :  deriving POA from a fixed POA/EC ratio is uncertain, even more since the used ratio has been derived some 20 years ago. No control is possible using daily measurements.  This is in line with my above remark that the used observational data-set is not suitable for in depth model evaluation.*
**Response:** We agree with the reviewer. As discussed in the responses to comment 2 and 3, CAMx-UNIPAR will be evaluated in various domains and periods using different gas mechanism such as CB6.

***Comment 11***
*Line 270: it is not specifically figure "Fig. 4(e–h)" here.*
**Response:** Fig. 4(e-h) is correct for the time series of the simulated SOA species and pie charts.

***Comment 12***
*Line 274 : "For example, the fraction of alkane SOA was higher than that of terpene SOA in all sites except Sacramento." Is this SOA from the ALK5 lumped species? Does this include long chain alkanes, of intermediate volatility. Please state this before the conclusion section.*
**Response:** Alkane SOA includes long chain alkane (up to C24). As discussed in the response to comment 5, the description of alkane has been modified in the revised manuscript (Section 2.2).

***Comment 13***
*Lines 410 – 427: there is a long discussion on the potential NOx and SO₂ reduction impact on BSOA formation. I encourage the authors to add a test simulation where they test such scenarios over their model domain.*
**Response.** We agree to the reviewer. In the future, we will simulate the impact of acidity on SOA formation with better emission data in the different locations. The sensitivity of SOA formation to $SO_2$ emissions is complex because the increased acidity due to increased $SO_2$ can reduce ammonium nitrate, which is hygroscopic and influences aqueous reactions. In this paper, the impact of $NO_x$ on SOA formation has been discussed to response to the reviewer. Section 3.5 "Impact of $NO_2$ on SOA formation" in "Results and Discussion" has been added into the revised manuscript. The newly added Table 3 shows the correlation coefficients between $NO_2$, nitrate and the SOA mass associated with each HC excluding low PBL height periods. Based on the formaldehyde-to-$NO_2$ ratio (FNR) at Figure S14, the four urban sites of this study are VOC-limited condition (high $NO_x$ levels). Under the VOC-limited condition, SOA production is typically correlated to $NO_2$ with a negative relation. However, SOA simulations of this study are positively correlated to $NO_2$. Biogenic SOA is more strongly correlated with $NO_2$ than anthropogenic SOA. This information has been discussed in the revised paper (Section 3.5 and Fig. S14) and reads now,

"3.5 Impact of $NO_2$ on SOA formation

To understand the impact of $NO_2$ on SOA formation, the correlation coefficients (R) between $NO_2$, nitrate, and SOA mass associated with each HC were estimated as shown in Table 3. The simulations of $NO_2$, nitrate and SOA during low PBL height periods were excluded. The formaldehyde-to-$NO_2$ ratio (FNR) is typically used to denote $NO_x$ levels. When FNR is less than 1, it represents VOC-limited condition (Duncan et al., 2010; Hoque et al., 2022). Based on the spatial distribution of FNR in Fig. S14, the four urban sites of this study were VOC-limited (high $NO_x$ levels) condition. Under this environment, a typical SOA production in daytime is negatively correlated with $NO_2$ concentrations (Presto et al., 2005; Yang et al., 2020; Madhu et al., 2022). However, all correlation coefficients between $NO_2$ and SOA mass shown in Table 3 were positive. Biogenic SOA was more strongly correlated with $NO_2$ (larger positive R) than anthropogenic SOA.

As discussed in Section 3.2 and 3.3, biogenic HCs, particularly terpene, react with a nitrate radical to form low-volatile products and effectively increase SOA mass. $NO_2$ is linked to the formation of nitrate radicals at night and thus, it can be positively related to biogenic SOA mass. Increased nighttime biogenic SOA (Fig. 5) can influence OMP of anthropogenic SOA and it can weaken the negative correlation between $NO_2$ and anthropogenic SOA formation. For example, the last column in Fig. 5, which displays the diurnal variation of the total SOA concentrations, evinces the influence of terpene SOA on anthropogenic SOA mass. Anthropogenic SOA mass gradually increased with increasing terpene SOA mass after sunset (5 PM) when the production of OH radical was nearly inactive. As shown in Fig. S15, a rapid change in the PBL height appeared between 3PM and 5PM, and no change appeared in the PBL height after 7PM. Thus, the influence of the lowered PBL height on changes in anthropogenic SOA mass after 7PM can be excluded. Under the Central Valley's environments in this study, ammonia was rich, temperature was relatively low, and humidity was high. The high concentration of $NO_2$ in this region is favorable to form inorganic nitrate aerosol, which can promote aqueous phase reactions of organic species. Evidently, a strong positive correlation appeared between inorganic nitrate concentrations and SOA mass (R: 0.41–0.86 in Table 3) declining a conventional negative correlation between $NO_x$ concentration and anthropogenic SOA formation at high $NO_x$ levels."

In addition, we have added the description about the $NO_x$ and $SO_2$ impacts on SOA mass into Section 4 in the revised manuscript, and reads now,

"As discussed in Section 3.5, the anthropogenic SOA formation can be influenced by the amount of biogenic SOA, and vice versa. Hence the impact of $NO_2$ on SOA formation can be varied with the composition of precursor HCs. The SOA mass in California urban area possibly would decrease as $NO_x$ decreases because of the dominance of

terpene SOA, which has a strongly positive correlation with $NO_2$ as shown in Table 3. For the polluted urban area where anthropogenic SOA is dominant and $NO_x$ is rich, SOA formation can be negatively correlated with $NO_x$. Under this situation, SOA mass increases with lowering $NO_x$. The impact of $SO_2$ change on SOA formation is complicated due to the aerosol acidity and the amount of wet-inorganic aerosol. The reduction of $SO_2$ drops aerosol acidity and thus, it can reduce SOA mass. However, the reduced aerosol acidity with decreasing $SO_2$ under ammonia-rich environments can increase the deposition of nitric acid forming ammonium nitrate that is very hygroscopic. The ERH of ammonium nitrate is lower than that of ammonium sulfate and increases wettability of aerosol. Increased ammonium nitrate mass increases partitioning of polar organics onto the wet aerosol and enhances the reactions of reactive organic species in aqueous phase."

**Comment 14**
*In table 1, I guess that SOA data means chamber data for validation of parts of the UNIPAR model. Please correct. I do not understand why 50 *4 stoechiometric coefficients are needed.*
**Response:** Yes, SOA data is chamber data. The word "SOA data" has been replaced by "Chamber data". We added information into the footnote "c" in Table 1. The description of four different stoichiometric coefficient sets is also described in footnote "b" in Table 1.

**Comment 15**
*For regional simulations, I think Yu et al. 2022 should appear more often except for alkanes.*
**Response:** We have cited the paper by Yu et al. (2022) in numerous places of this manuscript and Table 1.

**Comment 16**
*Line 45: "The SOA model has been simulated in regional and global scales ......" This is a strange formulation. You could say : "the ....model has been to simulate .... at regional and global scales …."*
**Response:** "in regional and global scales" has been replaced by "at regional and global scales"

**Comment 17**
*Line 69: "established on UNIPAR »   in, within?*
**Response:** "The model parameters and equations established on UNIPAR" has been replaced by "The model parameters and equations in UNIPAR"

**Comment 18**
*Line 99 : "as follow » with an « s ».*
**Response:** "as follow" has been replaced by "as follows"

**Comment 19**
*Line 101 : "For the SOA formation in multiphase ..." -> "For the multiphase SOA formation ..."*
**Response:** "For the SOA formation in multiphase" has been replaced by "For the multiphase SOA formation"

**Comment 20**
*Line 107 : " volatility-reactivity base » -> « volatility-reactivity based"*
**Response:** "volatility-reactivity base" has been replaced by "volatility-reactivity based"

**Comment 21**
*Line 115 : "are used to manage their multiphase partitioning"  -> may be : "are used to determine their multiphase partitioning"*
**Response:** "are used to manage their multiphase partitioning" has been replaced by "are used to determine their multiphase partitioning"

**Comment 22**
*Line 422: "where aromatic SOA is MORE dominant than that in California"*
**Response:** This sentence has been removed in the revised manuscript.

**Comment 23**
*Line 436: "can yield lower SOA yields" please reformulate*
**Response:** "can yield" has been replaced by "have"

**References**

[revised manuscript text omitted]

---

## Author Comment (AC2)

**Manuscript #: egusphere-2023-93**

**Response to Anonymous Referee #2**

We would like to thank the reviewer for the time and the constructive comments on our work. The comments are reproduced below along with the author response. The changes made to the manuscript or supporting information were in red color.

*Comment 1*
*In the introduction reference is made to "the SOA model" and I'm an unclear if this referring to a particular model or the more general idea of modelling SOA. This might be a case of just refining the language.*
**Response:** The word "the SOA model" in Section 1 has been replaced by "SOA models".

*Comment 2*
*"Volatility-reactivity based lumping species originating from explicit gas mechanisms allow to estimate their physicochemical parameters that process multiphase partitioning and in-particle reactions (i.e., oligomerization and acid catalyzed reactions)." I'm not sure I fully understand this sentence. Do you mean that by adopting an approach where species from a gas phase chemical mechanism are lumped based on their volatility and reactivity allows their physicochemical parameters, which are highly influential for the partitioning to multiphase aerosol and the subsequent in-particle reactions to be estimated?*
**Response:** To response to the reviewer, the sentence has been revised as below,

"Chemical species originating from explicit gas mechanisms were lumped based on volatility and reactivity in the aerosol phase. The physicochemical parameters of lumping species allow the UNIPAR model to process multiphase partitioning and in-particle reactions (i.e., oligomerization and acid catalyzed reactions)."

*Comment 3*
*I don't understand the use of the word "emerging" in line 56 – please could you clarify?*
**Response:** The word "emerging" has been removed.

*Comment 4*
*I think it would be worth mentioning that uncertainty in emissions of primary OM and secondary OM precursors may also contribute to model biases.*
**Response:** Please find Section "3.1 Simulation of organic matter" for the description of the uncertainty in primary OC simulation and observation. For the secondary SOA precursors, please find the last paragraph in Section "4 Atmospheric implication and uncertainties" in the revised manuscript.

*Comment 5*
*In the final paragraph you mention the expansion of the CAMx–UNIPAR model to include alkane SOA and nighttime chemistry of biogenic HCs in this work. There are then further descriptions of other changes made to CAMx–UNIPAR in other studies. It is not clear if these changes are also included in the version of CAMx–UNIPAR used in this study or not. This needs to be clarified. It would also be helpful if the additions to specific to this study were described in more detail and listed in the SI.*
**Response:** To clearly show the new additions in the CAMx-UNIPAR model of this study, the last paragraph of Section 1 has been revised as follows,

"In this study, the CAMx–UNIPAR model has been updated to include the SOA formation from long-chain alkanes and nighttime chemistry of biogenic HCs. Long-chain alkanes are regarded as essential precursors for SOA formation (Aumont et al., 2012; Madhu et al., 2022). Madhu et al. (2022) have recently added an autoxidation mechanism into alkane semi-explicit oxidation mechanisms, improving the predictability of alkane SOA using the UNIPAR model against their chamber study. The resulting alkane model parameters have been newly implemented into CAMx-UNIPAR. In addition, the UNIPAR model of this study has been expanded to simulate biogenic SOA based on three major oxidation paths (i.e., OH radicals, ozone and nitrate radicals) being capable of nighttime SOA formation that is dominated by oxidation with ozone and nitrate radicals (Han and Jang,

2023)."

**Comment 6**

*"These physicochemical parameters are universalized for five major precursor groups in UNIPAR (Table 1)." What does this mean?*

**Response:** The sentence has been revised to clearly explain what five major precursor groups are as follows,

"These physicochemical parameters are unified for five major precursor groups (aromatics, alkanes, terpene, sesquiterpene, and isoprene) as shown in Table 1."

**Comment 7**

*Where are SOA precursor emissions coming from? You mention they are "SAPRC07-based" but I don't know what this means.*

**Response:** The sentence has been revised and reads now,

"The emission data of HCs is speciated to use SAPRC07 gas mechanisms...."

**Comment 8**

*Have the emissions been validated? A bias in these emissions could mean the SOA model is getting the right/wrong answers for the wrong reasons in some cases. Fig 3 suggests a low bias in SOA which might come in part from an emissions bias.*

**Response:** The emission data used in this study are based on California ARB(Air Resources Board)'s regional inventories and their own estimates and emission monitoring data for local sources. The emission processing went through QA/QC procedures.

**Comment 9**

*"The lumping species in the lowest volatility is treated as non-volatile OM in this study." I don't understand this sentence – do you mean that the lumped species with the lowest volatility is automatically treated as non-volatile OM and so irreversibly partitions into the aerosol phase?*

**Response:** The additional description has been added into Section 2.1 in the revised manuscript and reads now,

"The lumping species in the lowest volatility group are involved in oligomerization with a high reaction rate constant used for glyoxal, regardless of lumping groups' reactivity scale. The species in the lowest volatility group, which are multifunctional and dominantly present in aerosol phase, easily react in aerosol phase via various unidentified reactions (esterification and oxidations) and form non-volatile species."

**Comment 10**

*In terms of OMP and OMH, I am unclear which relates to volatility-driven partitioning, which relates to reactive uptake and which relates to the dissolution of gases in aqueous phase aerosol. Differentiating between these is a key part of this model and so more detail is needed here.*

**Response:** In order to clearly describe the UNIPAR model, the description of OMP and OMH in Section 2.1 has been modified in the revised manuscript as follows,

"4) The concentration of lumping species is distributed into gas ($C_g$), organic ($C_{or}$), and inorganic phases ($C_{in}$) using partitioning coefficients estimated based on Pankow's absorptive partitioning model (Pankow, 1994) with vapor pressure, the estimated activity coefficients of lumping species in both the organic and inorganic phases (Zhou et al., 2019; Yu et al., 2021c; Han and Jang, 2022; Madhu et al., 2022; Han and Jang, 2023), and aerosol's average molecular weight in each phase.

5) Kinetic parameters, such as lumping species' reactivity scales and their basicity constants, to calculate aerosol phase reaction rate constants in the organic phase and inorganic phase are reported in Tables S4–S5, respectively. The kinetic parameters used in CAMx-UNIPAR are updated by removing the artifact from gas-wall partitioning (Han and Jang, 2020; Han and Jang, 2022)

6) Heterogeneously formed OM (OMH), which is produced via oligomerization in the organic phase and the inorganic phase, is treated as non-volatile OM. The impact of viscosity on aerosol growth is also considered by including the equation term as a function of the average molecular weight of OM and the O:C ratio (Han and Jang,

2022). Aqueous reactions in the presence of wet-inorganic aerosol are operated by acid-catalyzed reactions and organosulfate formation and are processed under broad ranges of aerosol acidity ([H+]) and relative humidity (RH) levels to form both dry and wet inorganic aerosols.   The lumping species in the lowest volatility group are involved in oligomerization with a high reaction rate constant used for glyoxal, regardless of lumping groups' reactivity scale. The species in the lowest volatility group, which are multifunctional and dominantly present in aerosol phase, easily react in aerosol phase via various unidentified reactions (esterification and oxidations) and form non-volatile species.

7) The SOA mass in UNIPAR is estimated by gas-particle partitioning (OMP) and heterogeneous reactions (OMH) in both organic and inorganic phase. The SOA mass formed from partitioning (OMP) is estimated using the Newtonian method (Schell et al., 2001) based on a mass balance of organic compounds between the gas and particle phases governed by Raoult's law. OMH is considered as a pre-existing absorbing organic material for gas-particle partitioning (Cao and Jang, 2010; Im et al., 2014). The resulting OMP is updated by the addition of $C_{in}$."

*Comment 11*
*Line 245 – suggest you replace "degradation" with "decrease"*
**Response:** "degradation" has been replaced by "decrease".

*Comment 12*
*Throughout this paragraph I would use "bias" in place of "deviation".   For example, "The low bias of the predicted SOA is generally greater than the high bias of the POA, which drives the low bias of the total OM from the observations."*
**Response:** "deviation" in Section 3.1 has been replaced by "bias"

*Comment 13*
*"The underestimation of SOA mass can be attributed to missing precursor HCs and unidentified chemistry in the gas and aerosol phases. For example, the SOA model is currently missing phenols, branched and cyclic alkanes, and polyaromatic hydrocarbons (i.e., naphthalene)."   I'm not clear this is true without an evaluation of the emissions of the precursor species.*
**Response:** The parameters in UNIAPR are continuously updated with the better gas mechanisms and expanded to include missing hydrocarbons. To support this information, we added citations as follows,

"For example, the precursor HCs such as phenols (Bruns et al., 2016; Majdi et al., 2019; Choi and Jang, 2022), branched and cyclic alkanes (Chan et al., 2013; Gentner et al., 2017; Madhu et al., 2023), and polyaromatic hydrocarbons (i.e., naphthalene) (Riva et al., 2015; Wang et al., 2021) are currently missing in the UNIPAR model."

*Comment 14*
*"The simulated SOA/POA ratios were relatively lower than those in the observed ratios, as discussed for the different deviations of the predicted POA and SOA from the observations." I'm not sure I understand the second clause here. Do you mean that the lower SOA/POA ratio from the model is in line with the general model high bias of POA and low bias of SOA?*
**Response:** The sentence has been revised as follows,

"The simulated SOA/POA ratios were relatively lower than those in the observed ratios, which are calculated using decoupled SOA and POA with a POC/EC ratio (Sect. 2.4), suggesting that POA is overpredicted and SOA is underpredicted in the CAMx-UNIPAR simulations."

*Comment 15*
*"A strong wind appeared in the northern area, decreasing the residence time of pollutants, which reduced secondary products of pollutants in this region." What time period are you discussing here? From Fig S3, I can see wind speeds at San Jose are persistently higher than for the other locations.*
**Response:** Overall, winds in the northern area were strong. To clarify this, "during the simulation period (Table S7 and Fig. S4)" has been added as below,

"A strong wind appeared in the northern area during the simulation period (Table S7 and Fig. S4), decreasing the

*residence time of pollutants, which reduced secondary products of pollutants in this region."*

***Comment 16***

*I understand that you cannot do anything about the 3-day averaging of the observational data but I do think the resulting lower number of observation data points means extending these simulations for at least another month or two would be warranted.*

**Response:** We agree with the reviewer. The simulation of the air quality in CA was the first case to demonstrate CAMx-UNIPAR to regional simulation in the USA domain. Due to the frequent occurrence of wildfire in California, we had no choice but to select a season (Jan-Feb), when wildfires were inactive in 2018. In the future, CAMx-UNIPAR will be evaluated in different regions and different time.

***Comment 17***

*It would also be helpful to have a timeseries plot of emissions at each site in a similar format to the line plots of Fig 3.*

**Response:** Timeseries plots of emissions at each site have been added in the revised SI (Fig. S9)

***Comment 18***

*In Figure 2(b-h), you give units of moles or g per second. I think the units of emission should be moles or g per second per unit area (I admit the final column in (b) could stay as moles/s). While I understand that the magnitude of the emissions of the different species vary considerably such that it would not be sensible to have a single common colorbar range for c-h, could cleanly separated ticks (e.g. 0, 2, 4, 6, 8, 10) be used for each to make comparison easier?*

**Response:** Fig. 2 has been updated with a larger font size and the unit at mole (or g) $s^{-1}$ $grid^{-1}$.

***Comment 19***

*Similarly for Fig 6, I would strongly encourage the authors to consider either a log scale for the colorbar or make it much clearer that the concentrations span 2 orders of magnitude.*

**Response:** Fig. 6 has been updated with a larger font size.

***Comment 20***

*"Additionally, the model includes the low volatility products originating from autoxidation of α-pinene ozonolysis products (Roldin et al., 2019; Crounse et al., 2013; Bianchi et al., 2019). The importance of autoxidation mechanisms on terpene SOA formation was in a recent study by Yu et al. (2021c) for the daytime chemistry (Yu et al., 2021c)." While it is very good that you are considering highly oxidized species from α-pinene, could you provide any information about the yields you are using or whether you are using the full Roldin scheme which is substantial. Furthermore, the reference to the paper by Yu et al (2021c) is vague – what did this paper show?*

**Response:** Additional description has been added as below,

"The importance of autoxidation mechanisms on terpene SOA formation was demonstrated in a recent study by Yu et al. (2021c) for daytime chemistry. In their study, the peroxy radical autoxidation mechanism (PRAM) developed by Roldin et al. (2019) was included in UNIPAR to evaluate the impact of HOMs on terpene SOA formation. In the sensitivity test of SOA prediction associated with PRAM, α-pinene SOA mass increased by 15–35% in the presence of PRAM, suggesting that substantial impact of PRAM on the total SOA mass (Yu et al., 2021c)."

***Comment 21***

*Unless I have misunderstood the difference between OMH and OMP, I would have thought that the alkane autoxidation products would be in the OMP category given their highly oxidized structure and low volatility – please could you clarify?*

**Response:** Some autoxidation products from alkanes belong to the lowest volatility group and participate in heterogeneous reactions to form OMH. Other autoxidation products, which are classified into partitioning only or slow reactivity are involved in OMP. Please find the response to comment 10.

***Comment 22***

*A better color scale is needed for O₃ in Fig S10 since most of the region is off the top end of the scale.*

**Response:** Fig. S10 in the original manuscript has been changed to Fig. S11 in the revised manuscript with a different color scale for $O_3$.

***Comment 23***
*I am surprised by the low yield of SOA from aromatics. Can you provide any more detail about why this is quite so low?*
**Response:** The low contribution of aromatic SOA to the total SOA mass is related to emissions of precursor HCs. Alkane is a dominant in anthropogenic HC emission (Fig. 2). Please also see the response to comment 5 from reviewer 1.

***Comment 24***
*Data Accessibility. In the interests of community modelling and FAIR principles, I would like to see the code and model data uploaded to a freely accessible repository such as Github or Zenodo.*
**Response:** The code of the CAMx-UNIPAR model is available upon request in Github.
CAMx-UNIPAR ver. 1.1, which included aromatics and biogenic daytime, is available in GitHub.   The updated CAMx-UNIPAR ver. 1.2 has been preparing to include various precursors including alkanes (linear and branched alkanes), updated aromatics, and day/night biogenics.   The updated version will include more hydrocarbons than the simulation of this study.

**References**

[revised manuscript text omitted]

---

## Referee Report (RR1)

**Review of Jo et al (2023)**

I have reviewed the authors' response to my original comments, and I thank them for largely addressing my concerns. However, I believe further work is required to address fully the concerns raised in the following comments. For completeness, I include my original comment, the authors' response, and my subsequent response.

**Comment 6**

*"These physicochemical parameters are universalized for five major precursor groups in UNIPAR (Table 1)."*
*What does this mean?*
**Response:** The sentence has been revised to clearly explain what five major precursor groups are as follows,

"These physicochemical parameters are unified for five major precursor groups (aromatics, alkanes, terpene, sesquiterpene, and isoprene) as shown in Table 1."

**Reviewer response:**
While you have changed "universalized" to "unified", it is still unclear to me what this means. Do you mean that physicochemical parameters are the same for aromatics, alkanes, terpene, sesquiterpene, and isoprene? This appears to be the case looking at Table 1 (although I note the value is 51 for isoprene and 50 for all the others). Could you say the "physicochemical parameters are the same for the five major precursor groups"?

**Comment 8**

*Have the emissions been validated? A bias in these emissions could mean the SOA model is getting the right/wrong answers for the wrong reasons in some cases. Fig 3 suggests a low bias in SOA which might come in part from an emissions bias.*
**Response:** The emission data used in this study are based on California ARB(Air Resources Board)'s regional inventories and their own estimates and emission monitoring data for local sources. The emission processing went through QA/QC procedures.

**Reviewer response:**
It is good to see the authors are using a validated emission inventory. The use of the California ARB's regional inventory and a citation to its documentation and evaluation should be included in the main text.

**Comment 13**

*"The underestimation of SOA mass can be attributed to missing precursor HCs and unidentified chemistry in the gas and aerosol phases. For example, the SOA model is currently missing phenols, branched and cyclic alkanes, and polyaromatic hydrocarbons (i.e., naphthalene)." I'm not clear this is true without an evaluation of the emissions of the precursor species.*

**Response:** The parameters in UNIAPR are continuously updated with the better gas mechanisms and expanded to include missing hydrocarbons. To support this information, we added citations as follows,

"For example, the precursor HCs such as phenols (Bruns et al., 2016; Majdi et al., 2019; Choi and Jang, 2022), branched and cyclic alkanes (Chan et al., 2013; Gentner et al., 2017; Madhu et al., 2023), and polyaromatic hydrocarbons (i.e., naphthalene) (Riva et al., 2015; Wang et al., 2021) are currently missing in the UNIPAR model."

**Reviewer response:**
I don't view this as an acceptable response, and it does not address my concerns. I understand that UNIPAR undergoes constant development (and that is a good thing) but that is not the issue at hand here. The issue at hand is whether the omission of these key species is a major driver of the SOA low bias or whether the low bias is being caused by deficiency in the model elsewhere.

A range of citations have been given but no context provided, and this is insufficient. It is the job of the author to provide convincing evidence to support their claim. These papers may refer to the SOA yield from the various omitted species, but it is not clear. The authors should highlight key features of these papers both in their response to the reviewers, and in the main text for the benefit of other readers. They should make it clear if these omitted species do indeed produce SOA at appreciable yields.

One of UNIPAR's key aims to simulate SOA well so drivers of its biases must be explored in detail. To fully support their claim that the underestimation of SOA is attributable to these missing precursors, rather than another deficiency in the mechanism, the authors should estimate the production of SOA from these missing species (regional emissions times approximate SOA yield) and show how this extra SOA production term might help resolve the model low bias by putting it into context with the SOA production currently included in this version of UNIPAR for the simulations considered.

***Comment 24***
*Data Accessibility. In the interests of community modelling and FAIR principles, I would like to see the code and model data uploaded to a freely accessible repository such as Github or Zenodo.*

**Response:** The code of the CAMx-UNIPAR model is available upon request in Github.
CAMx-UNIPAR ver. 1.1, which included aromatics and biogenic daytime, is available in GitHub. The updated CAMx-UNIPAR ver. 1.2 has been preparing to include various precursors including alkanes (linear and branched alkanes), updated aromatics, and day/night biogenics. The updated version will include more hydrocarbons than the simulation of this study.

**Reviewer response:**
This response has only partially answered my original comment. The code of the CAMx-UNIPAR should be freely available on GitHub (not available on request) and the web address clearly provided.

There is also no comment on the availability on model data. This must be made available on a repository such as Zenodo and clearly signposted in the text so as to be inline with Copernicus' data accessibility policy.

---

## Author Response (AR2)

**Manuscript #: egusphere-2023-93**

**Response to Anonymous Referee #2**

We would like to extend our appreciation to the reviewer for the time and constructive comments. The comments are reproduced below along with the author response. The changes made to the manuscript were in red color.

**Comment 6**

*"These physicochemical parameters are universalized for five major precursor groups in UNIPAR (Table 1)."*
*What does this mean?*
**Response:** The sentence has been revised to clearly explain what five major precursor groups are as follows,

"These physicochemical parameters are unified for five major precursor groups (aromatics, alkanes, terpene, sesquiterpene, and isoprene) as shown in Table 1."

*Reviewer response:*
*While you have changed "universalized" to "unified", it is still unclear to me what this means. Do you mean that physicochemical parameters are the same for aromatics, alkanes, terpene, sesquiterpene, and isoprene? This appears to be the case looking at Table 1 (although I note the value is 51 for isoprene and 50 for all the others). Could you say the "physicochemical parameters are the same for the five major precursor groups"?*
**Response:** The sentence has been revised as follows,
"Each precursor group uses a single set of physicochemical parameter arrays associated with lumping species: for example, 50 arrays for aromatics; 50 arrays for alkanes; 50 arrays for terpene; 50 arrays for sesquiterpene; and 51 arrays for isoprene."

**Comment 8**

*Have the emissions been validated? A bias in these emissions could mean the SOA model is getting the right/wrong answers for the wrong reasons in some cases. Fig 3 suggests a low bias in SOA which might come in part from an emissions bias.*
**Response:** The emission data used in this study are based on California ARB(Air Resources Board)'s regional inventories and their own estimates and emission monitoring data for local sources. The emission processing went through QA/QC procedures.

*Reviewer response:*
*It is good to see the authors are using a validated emission inventory. The use of the California ARB's regional inventory and a citation to its documentation and evaluation should be included in the main text.*
**Response:** The following sentences were added to the Section 2.2 and the Acknowledgments section.
"Emission inputs are based on California Air Resources Board regional inventories and provided by the Bay Area Air Quality Management District (BAAQMD, 2023)." (Sect. 2.2). The website address for emission inputs has been included in the reference in the revised manuscript.
"Emissions and meteorological inputs for the 2018 California regional modeling were provided by the Bay Area Air Quality Management District." (Acknowledgments section)

**Comment 13**

*"The underestimation of SOA mass can be attributed to missing precursor HCs and unidentified chemistry in the gas and aerosol phases. For example, the SOA model is currently missing phenols, branched and cyclic alkanes, and polyaromatic hydrocarbons (i.e., naphthalene)."*
*Original comment: I'm not clear this is true without an evaluation of the emissions of the precursor species.*
**Response:** The parameters in UNIAPR are continuously updated with the better gas mechanisms and expanded to include missing hydrocarbons. To support this information, we added citations as follows,

"For example, the precursor HCs such as phenols (Bruns et al., 2016; Majdi et al., 2019; Choi and Jang, 2022),

branched and cyclic alkanes (Chan et al., 2013; Gentner et al., 2017; Madhu et al., 2023), and polyaromatic hydrocarbons (i.e., naphthalene) (Riva et al., 2015; Wang et al., 2021) are currently missing in the UNIPAR model."

*Reviewer response:*

*I don't view this as an acceptable response, and it does not address my concerns. I understand that UNIPAR undergoes constant development (and that is a good thing) but that is not the issue at hand here. The issue at hand is whether the omission of these key species is a major driver of the SOA low bias or whether the low bias is being caused by deficiency in the model elsewhere.*

*A range of citations have been given but no context provided, and this is insufficient. It is the job of the author to provide convincing evidence to support their claim. These papers may refer to the SOA yield from the various omitted species, but it is not clear. The authors should highlight key features of these papers both in their response to the reviewers, and in the main text for the benefit of other readers. They should make it clear if these omitted species do indeed produce SOA at appreciable yields.*

*One of UNIPAR's key aims to simulate SOA well so drivers of its biases must be explored in detail. To fully support their claim that the underestimation of SOA is attributable to these missing precursors, rather than another deficiency in the mechanism, the authors should estimate the production of SOA from these missing species (regional emissions times approximate SOA yield) and show how this extra SOA production term might help resolve the model low bias by putting it into context with the SOA production currently included in this version of UNIPAR for the simulations considered.*

**Response:** It is difficult to estimate the SOA mass associated with missing hydrocarbons in the regional scales without model parameters for each precursor. Most laboratory studies report SOA yields from the oxidation of hydrocarbons at high concentrations, and focus on aerosol characterization without parameterization for missing hydrocarbons. The sentence that was commented by the reviewer has been revised and this reads now,

"The precursor HCs such as phenols (Bruns et al., 2016; Majdi et al., 2019; Choi and Jang, 2022), branched and cyclic alkanes (Chan et al., 2013; Gentner et al., 2017; Madhu et al., 2023), and polycyclic aromatic HCs (PAHs) (i.e., naphthalene) (Riva et al., 2015; Wang et al., 2021) are currently missing in the UNIPAR model. For example, all alkanes in this study are treated with linear alkanes, increasing some uncertainties. Branched alkane SOA can be slightly overestimated by substituting it with linear alkanes at the same carbon length, but cyclic alkanes can be underestimated by using linear alkanes (Madhu et al., 2022; Madhu et al., 2023). Zhang and Ying (2012) reported that PAHs including naphthalene, methylnaphthalene and dimethylnaphthalene can contribute to 4% of the anthropogenic SOA mass. Pye et al. (2022) reported in their regional simulation using the Community Regional Atmospheric Chemistry Multiphase Mechanism (CRACMM) that a significant amount of phenolic compounds is missing in the current model. They estimated that most phenol mass (69%) is directly emitted with the balance from benzene oxidation, and cresols are mainly chemically produced (80%). The missing emissions of phenol and cresol can account for 30% of the total aromatic SOA mass (Pye et al., 2022). The SOA formation from low volatility aromatic HCs (aromatics substituted with long-chain alkyl groups) is also missing in the SOA simulation of this study."

**Comment 24**

*Data Accessibility. In the interests of community modelling and FAIR principles, I would like to see the code and model data uploaded to a freely accessible repository such as Github or Zenodo.*

**Response:** The code of the CAMx-UNIPAR model is available upon request in Github.
CAMx-UNIPAR ver. 1.1, which included aromatics and biogenic daytime, is available in GitHub. The updated CAMx-UNIPAR ver. 1.2 has been preparing to include various precursors including alkanes (linear and branched alkanes), updated aromatics, and day/night biogenics. The updated version will include more hydrocarbons than the simulation of this study.

*Reviewer response:*

This response has only partially answered my original comment. The code of the CAMx-UNIPAR should be freely available on GitHub (not available on request) and the web address clearly provided.
There is also no comment on the availability on model data. This must be made available on a repository such as Zenodo and clearly signposted in the text so as to be inline with Copernicus' data accessibility policy.

**Response:** The code of CAMx-UNIPAR and model data (meteorological and emission inputs) will be provided upon request. The code of CAMx-UNIPAR has been updating to include more precursors, and the user manual

needs to be prepared.   When CAMx-UNIPAR is ready for the manual and updated parameters for essential precursors, CAMx-UNIPAR will be freely available for publics. In current, code to run the CAMx-UNIPAR model in this study is available upon request with appropriate purpose. In addition, the use of input data needs permission from BAAQMD.

**References**

Bay Area Air Quality Management District (BAAQMD), https://www.baaqmd.gov/, last access: 2 November 2023.

Bruns, E. A., El Haddad, I., Slowik, J. G., Kilic, D., Klein, F., Baltensperger, U., and Prevot, A. S.: Identification of significant precursor gases of secondary organic aerosols from residential wood combustion, Sci. Rep., 6, 27881, 10.1038/srep27881, 2016.

Chan, A. W. H., Isaacman, G., Wilson, K. R., Worton, D. R., Ruehl, C. R., Nah, T., Gentner, D. R., Dallmann, T. R., Kirchstetter, T. W., Harley, R. A., Gilman, J. B., Kuster, W. C., de Gouw, J. A., Offenberg, J. H., Kleindienst, T. E., Lin, Y. H., Rubitschun, C. L., Surratt, J. D., Hayes, P. L., Jimenez, J. L., and Goldstein, A. H.: Detailed chemical characterization of unresolved complex mixtures in atmospheric organics: Insights into emission sources, atmospheric processing, and secondary organic aerosol formation, J. Geophys. Res.-Atmos., 118, 6783-6796, 10.1002/jgrd.50533, 2013.

Choi, J. and Jang, M.: Suppression of the phenolic SOA formation in the presence of electrolytic inorganic seed, Sci. Total Environ., 851, 158082, https://doi.org/10.1016/j.scitotenv.2022.158082, 2022.

Gentner, D. R., Jathar, S. H., Gordon, T. D., Bahreini, R., Day, D. A., El Haddad, I., Hayes, P. L., Pieber, S. M., Platt, S. M., de Gouw, J., Goldstein, A. H., Harley, R. A., Jimenez, J. L., Prevot, A. S., and Robinson, A. L.: Review of Urban Secondary Organic Aerosol Formation from Gasoline and Diesel Motor Vehicle Emissions, Environ. Sci. Technol., 51, 1074-1093, 10.1021/acs.est.6b04509, 2017.

Madhu, A., Jang, M., and Deacon, D.: Modeling the Influence of Chain Length on SOA Formation via Multiphase Reactions of Alkanes, EGUsphere, 2022, 1-30, https://doi.org/10.5194/egusphere-2022-681, 2022.

Madhu, A., Jang, M., and Jo, Y.: Modeling the influence of carbon branching structure on SOA formation via multiphase reactions of alkanes, EGUsphere, 2023, 1-28, 10.5194/egusphere-2023-1500, 2023.

Majdi, M., Sartelet, K., Lanzafame, G. M., Couvidat, F., Kim, Y., Chrit, M., and Turquety, S.: Precursors and formation of secondary organic aerosols from wildfires in the Euro-Mediterranean region, Atmos. Chem. Phys., 19, 5543-5569, 10.5194/acp-19-5543-2019, 2019.

Pye, H. O. T., Place, B. K., Murphy, B. N., Seltzer, K. M., D'Ambro, E. L., Allen, C., Piletic, I. R., Farrell, S., Schwantes, R. H., Coggon, M. M., Saunders, E., Xu, L., Sarwar, G., Hutzell, W. T., Foley, K. M., Pouliot, G., Bash, J., and Stockwell, W. R.: Linking gas, particulate, and toxic endpoints to air emissions in the Community Regional Atmospheric Chemistry Multiphase Mechanism (CRACMM) version 1.0, Atmos. Chem. Phys. Discuss., 2022, 1-88, https://doi.org/10.5194/acp-2022-695, 2022.

Riva, M., Robinson, E. S., Perraudin, E., Donahue, N. M., and Villenave, E.: Photochemical aging of secondary organic aerosols generated from the photooxidation of polycyclic aromatic hydrocarbons in the gas-phase, Environ. Sci. Technol., 49, 5407-5416, 10.1021/acs.est.5b00442, 2015.

Wang, X., Gemayel, R., Baboomian, V. J., Li, K., Boreave, A., Dubois, C., Tomaz, S., Perrier, S., Nizkorodov, S. A., and George, C.: Naphthalene-Derived Secondary Organic Aerosols Interfacial Photosensitizing Properties, Geophysical Research Letters, 48, 10.1029/2021gl093465, 2021.

Zhang, H. and Ying, Q.: Secondary organic aerosol from polycyclic aromatic hydrocarbons in Southeast Texas, Atmos. Environ., 55, 279-287, https://doi.org/10.1016/j.atmosenv.2012.03.043, 2012.